# AdaX: Adaptive Gradient Descent with Exponential Long Term Memory

## Abstract

Adaptive optimization algorithms such as RMSProp and Adam have fast convergence and smooth learning process. Despite their successes, they are proven to have non-convergence issue even in convex optimization problems as well as weak performance compared with the first order gradient methods such as stochastic gradient descent (SGD). Several other algorithms, for example AMSGrad and AdaShift, have been proposed to alleviate these issues but only minor effect has been observed. This paper further analyzes the performance of such algorithms in a non-convex setting by extending their non-convergence issue into a simple non-convex case and show that Adam's design of update steps would possibly lead the algorithm to local minimums. To address the above problems, we propose a novel adaptive gradient descent algorithm, named AdaX, which accumulates the long-term past gradient information exponentially. We prove the convergence of AdaX in both convex and non-convex settings. Extensive experiments show that AdaX outperforms Adam in various tasks of computer vision and natural language processing and can catch up with SGD.

## 1 Introduction

In the era of deep learning, Stochastic Gradient Descent (SGD), though proposed in the last century, remains the most effective algorithm in training deep neural networks (Robbins & Monro, 1951). Many methods have been created to accelerate the training process and boost the performance of SGD, such as momentum (Polyak, 1964) and Nesterov's acceleration (Nesterov, 1983). Recently, adaptive optimization methods have become popular as they adjust parameters' learning rates in different scales, instead of directly controlling the overall step size. These algorithms schedule the learning rates using a weighted average of the past gradients. For example, AdaGrad (Duchi et al., 2011) chooses the square root of the global average of the past gradients as the denominator of the adaptive learning rates. It is shown that when the gradients are sparse or small, AdaGrad can converge faster than vanilla SGD. However, its performance is quite limited in non-sparse settings.

Other adaptive algorithms have been proposed to replace the global average in AdaGrad by using the exponential moving average of past gradients, such as RMSProp (Tieleman & Hinton, 2012), AdaDelta (Zeiler, 2012), and Adam (Kingma & Ba, 2015). Among all these variants, Adam (Kingma & Ba, 2015) is the most popular yet controversial optimization algorithm since it has faster convergence rate than the others. However, Adam has worse performance (i.e. generalization ability in testing stage) compared with SGD. Recent theories (Wilson et al., 2017; Reddi et al., 2018) have also shown that Adam suffers from non-convergence issue and weak generalization ability. For example, Reddi et al. (2018) proposed that Adam's non-convergence problem originate from a mistake in their proof of convergence. They constructed a counterexample for Adam and thoroughly proved that Adam did not guarantee convergence even in a simple convex setting.

In the meantime, Zhou et al. (2019) theoretically proved that Adam could make a large update when gradients are small, and a small update when gradients are large, which probably lead the optimization process to wrong directions. Shazeer & Stern (2018) also empirically showed that Adam's parameter updates are not stable and its second moment could be out of date. Luo et al. (2019) examined the effective learning rate of Adam in training and found that Adam would produce too large or too small extreme learning rates. All the above analyses have suggested that Adam's unstable design of adaptive learning rate may impair the optimization process.

To address the above issues, we propose a new adaptive optimization method, termed AdaX, which guarantees convergence in both convex and non-convex settings. This is done by memorizing the long-term square of gradients exponentially as the second moment. To achieve this, we first extend the convex counterexample in Reddi et al. (2018) to a non-convex setting. We then analyze how the second moment instability in Adam and RMSProp could cause the optimization process to converge to a local minimum even without noisy gradients, revealing that Adam produces update steps that are too large in optimization. We also show how AMSGrad (Reddi et al., 2018) is unable to solve Adam's problem completely, because its effectiveness relies heavily on the magnitude of the maximal second moment, for instance, too large second moment would lead to insufficient training and thus sub-optimal solutions. To address the above problems, we introduce a novel AdaX algorithm and theoretically prove that it converges with a similar speed to Adam, but gets rid of the second moment instability. Extensive experiments show that AdaX outperforms Adam and is comparable to SGD with momentum in many tasks of computer vision and natural language processing.

## 2 BACKGROUND AND NOTATION

**Overview of Adaptive Methods.** To compare AdaX with the previous optimization methods, we follow (Reddi et al., 2018) to present a generic framework of adaptive algorithms as shown in Algorithm 1. Let $\mathcal{S}_d^+$ be a set of positive definite matrices in $\mathbb{R}^{d \times d}$, and $\mathcal{F}$ be the parameter domain. In line 1, we take $\phi_t : \mathcal{F} \to \mathbb{R}^d$ and $\psi_t : \mathcal{F} \to \mathcal{S}_+^d$ as input, which are unspecified moment functions that vary among different optimization algorithms. After obtaining the gradient at time $t$ in line 3, we can calculate the corresponding first and second moment $m_t, V_t$. In line 5, $\alpha_t = \alpha/\sqrt{t}$ is chosen as the step size in each iteration for the convergence analysis of adaptive algorithms. The projection operation in line 6, $\Pi_{\mathcal{F}, M}(y)$ is defined as $\operatorname{argmin}_{x \in \mathcal{F}} \| \sqrt{M}(x - y) \|$, where $M \in \mathcal{S}_d^+$ and $y \in \mathbb{R}^d$.

---
**Algorithm 1** Generic Adaptive Optimization Algorithm
---
1: **Input:** $x \in \mathcal{F}$, step size $\alpha$, sequence of functions $\{\phi_t, \psi_t\}_{t=1}^T$
2: **for** $t = 1$ **to** $T$ **do**
3:      $g_t = \nabla f_t(x_t)$
4:      $m_t = \phi_t(g_1, g_2, \ldots, g_t)$ and $V_t = \psi_t(g_1, g_2, \ldots, g_t)$
5:      $\alpha_t = \alpha/\sqrt{t}$
6:      $x_{t+1} = \Pi_{\mathcal{F}, \sqrt{V_t}}(x_t - \alpha_t m_t/\sqrt{V_t})$
7: **end for**
---

The main differences between the adaptive methods and the conventional SGD are in line 4 and 6, where the matrix $V_t$ scales the overall step size $\alpha_t$ element-wisely by $1/\sqrt{V_t}$, known as the adaptive learning rate. Using the general framework in Algorithm 1, we are able to summarize many adaptive optimization algorithms proposed recently. For example, AdaGrad (Duchi et al., 2011) designed its $V_t$ as the global average of past gradients, while Adam (Kingma & Ba, 2015) and RMSProp (Tieleman & Hinton, 2012) chose the exponential moving average as follows instead.

$$V_t = (\frac{1 - \beta_2}{1 - \beta_2^t})\operatorname{diag}(\sum_{i=1}^t \beta_2^{t-i} g_i^2), \qquad \text{(Adam)}$$

where $\beta_2$ is the fixed second moment coefficient and $g_i^2$ denotes the element-wise square of the gradients. The diagonal operation diag() perform the dimension transformation from $\mathbb{R}^d$ to $\mathcal{S}_+^d$. In order to improve the performance of Adam, Reddi et al. (2018) proposed AMSGrad, which took max operation on the second moment. Zhou et al. (2019) argued that we could replace $g_t^2$ in $V_t$ with some past gradient squares $g_{t-n}^2$ to temporarily remove the correlation between the first and second moment. Huang et al. (2019) constructed NosAdam, where a sequence of $\beta_{2t}$'s gave higher weights to past gradients. We provide a summary of different designs of adaptive learning rate in **Table 1**. It's noticeable that these algorithms, due to their exponential moving average design, still assign relatively high weights on recent gradients and past information is not emphasized. Besides, Loshchilov & Hutter (2019) noticed the difference between $L_2$ regularization and weight decay in adaptive algorithms and improved Adam with standard weight decay by proposing AdamW.

**Optimization Framework.** A commonly used framework for analyzing convex optimization algorithms was constructed by Zinkevich (2003), named the online optimization problem. In this

Table 1: Comparisons of different designs of the second moment

| | SGDM | AdaGrad | RMSProp |
|---|---|---|---|
| $\psi_t$ | $\mathbb{I}$ | $\text{diag}(\sum_{i=1}^t g_i^2/t)$ | $(1-\beta_2)\text{diag}(\sum_{i=1}^t \beta_2^{t-i} g_i^2)$ |
| | **Adam** | **AMSGrad** | **AdaShift** |
| $\psi_t$ | $(\frac{1-\beta_2}{1-\beta_2^t})\text{diag}(\sum_{i=1}^t \beta_2^{t-i} g_i^2)$ | $\text{diag}(\max(\hat{v}_{t-1}, v_t))$ | $\text{diag}(\beta_2 v_{t-1} + (1-\beta_2) g_{t-n}^2)$ |
| | **NosAdam** | **...** | **AdaX (ours)** |
| $\psi_t$ | $\text{diag}(\beta_{2t} v_{t-1} + (1-\beta_{2t}) g_{t-n}^2)$ | ... | $\frac{\beta_2}{(1+\beta_2)^t - 1}\text{diag}(\sum_{i=1}^t (1+\beta_2)^{t-i} g_i^2)$ |

framework setting, the optimization algorithm chooses a parameter set $\theta_t \in \mathcal{F}$ and an unknown cost function $f_t(\theta)$ evaluates its performance at $\theta_t$ in each iteration. Suppose that there exists a best parameter $f_t(\theta^*)$ such that $\theta^* = \text{argmin}_{\theta \in \mathcal{F}} \left( \sum_{t=1}^T f_t(\theta) \right)$, then a metric used to show the algorithm's performance is the regret function $R_T = \sum_{t=1}^T f_t(\theta_t) - f_t(\theta^*)$ and we want to ensure that $R_T = o(T)$ so that the algorithm will always converge to the optimal solution. (Zinkevich, 2003).

**Non-convergence of Adam.** Reddi et al. (2018) proposed that the matrix $\Gamma_t$ defined as follows, was mistakenly assumed to be positive semi-definite in the original convergence proof of Adam.

$$\Gamma_t = \left( \frac{\sqrt{V_{t+1}}}{\alpha_{t+1}} - \frac{\sqrt{V_t}}{\alpha_t} \right), \tag{1}$$

where $V_t$ and $\alpha_t$ are defined as in **Algorithm 1**. Based on such an observation, they constructed the following online convex optimization problem, in which Adam failed to converge to the optimal solution. Let $C > 2$ be a fixed constant and $\{f_t\}$ be the sequence of cost functions whose sum is to be minimized. Define $f_t$ as follows

$$f_t(x) = \begin{cases} Cx, \text{for } t \bmod 3 = 1 \\ -x, \text{otherwise} \end{cases} \tag{2}$$

where $x \in \mathcal{F} = [-1, 1]$. In this problem, Adam could not distinguish between the true large gradient direction ($C$) and the noisy gradient directions ($-1$) because its $\sqrt{V_t}$ scales the gradients to be of similar sizes, which forces the algorithm to reach a highly suboptimal solution $x = 1$ every three iterations. However, SGD and AdaGrad are both able to counteract the noisy gradients and converge to the optimum, which reveals the fact that Adam's design of adaptive learning rate is very unstable.

## 3    THE NONCONVERGENCE OF ADAM IN A NON-CONVEX SETTING

In this section, we extend the non-convergence problem in (2) to the non-convex setting and explain why the fast convergence of Adam impairs its performance in the long term. Let $C \in (1, +\infty), \lambda \in (0, 1)$ be constants in $\mathbb{R}$, consider the following simple sequence of non-convex functions $f_t$.

$$f_t(x) = \begin{cases} C\lambda^{t-1} x, \text{ for } x \geq 0 \\ \frac{C^2}{1-\lambda}, \text{ for } x < 0 \end{cases} \quad \forall t \geq 1 \tag{3}$$

It can be easily observed that for the domain $\mathcal{F} = [-2, C/(1-\lambda)]$, the minimum of each $f_t$ is obtained at $x = 0$. Suppose we start from $x_0 > 0$, then this problem simulates a situation where the gradient decreases exponentially as time increases, implying that the algorithm is approaching the global minimum and smaller step sizes are needed. Compared with the non-convergence problem proposed by Reddi et al. (2018), no high-frequency noise exist in our gradients. However, next to the optimal solution, there is a local minimum trap where no gradients exist and thus no algorithm could escape. Let $\alpha_1 = \alpha$ be the initial step size, we are able to show that SGD is capable of avoiding the trap, even without the learning rate decrease $\alpha_t = \alpha/\sqrt{t}$, and will converge to $x = 0$ if initialized well. However, Adam ignores the gradient decrease information and always enters the trap regardless of initialization. We summarize the above results in the following lemma

**Lemma 3.1** *In problem (3), with $\beta_1 = 0$, $\beta_2 \in (0, \lambda^2)$ and $\alpha_t \geq \alpha/t$, Adam will always reach the local minimum, i.e. $\sum_{t=1}^{T} f(t)/T \to \frac{C^2}{1-\lambda}$, $\forall x_1, \alpha_1 > 0$.*

We provide all the proofs in the Appendix. This parameter setting of Adam is the same as RMSProp except for the bias correction term, and the condition $\beta_1 < \sqrt{\beta_2}$ mentioned by Kingma & Ba (2015) is automatically satisfied. $\alpha_t \geq \alpha/t$ is a weak requirement for the step sizes and can be ensured with constant step sizes or $\alpha_t = \alpha/\sqrt{t}$ as in the convergence analysis. Intuitively, Adam would scale the decreasing gradient by $1/\sqrt{V_t}$, which approximately increases with the same speed. Therefore, its update steps would be larger than a fixed constant at any time step and would ultimately lead the parameters to the trap regardless of initialization. People may wonder whether the first moment design helps Adam in such a situation, but we can show that as long as the condition $\beta_1 < \sqrt{\beta_2}$ is satisfied, Adam would always reach the local minimum. Hence, Adam converges faster than SGD because of its large updates, but it cannot slow down when approaching the global minimum.

AMSGrad is constructed to address the problem of Adam's large steps during optimization. However, it suffers from two major issues in a non-convex situation. 1) The never decreasing $V_t$ in AMSGrad could lead to early stops and therefore insufficient training during optimization, as revealed by Huang et al. (2019); 2) The time for achieving the maximum of $V_t$ is uncontrollable. We show that for certain cases in our problem, AMSGrad is unable to help Adam.

**Lemma 3.2** *In problem (3), with $\beta_1 = 0$ and $\alpha_t \geq \alpha/t$, $\forall \beta_2 \in (0, 1), \exists \lambda \in (\sqrt{\beta_2}, 1)$, such that AMSGrad will always reach the local minimum, i.e. $\sum_{t=1}^{T} f(t)/T \to \frac{C^2}{1-\lambda}, \forall x_1, \alpha_1 > 0$.*

The lemma essentially states for any fixed $\beta_2$, we can find a $\lambda$ such that AMSGrad cannot help Adam because $V_t$ keeps increasing before stepping into the trap. Therefore, we still need an algorithm that can generate stable learning rates and control the update steps effectively. We emphasize that although the functions in (3) are not smooth, the problem does successfully provide some intuition on why Adam variants trains much faster than SGD, but cannot have comparable testing performance.

## 4 ALGORITHM AND CONVERGENCE

Next, we introduce our novel optimization algorithm and present its special way of adjusting the adaptive learning rate. Based upon the above discussions that current gradients lead to unstable second moment and that long-term memory algorithms are preferred, we design our algorithm AdaX by weighting exponentially more on the history gradients and less on the current gradients, as shown in **Algorithm 2**. The most important differences between AdaX and Adam are in line 6 and 7, where instead of an exponential moving average, we change $\beta_2$ to $1 + \beta_2$ and accumulate the past gradients. Such a design guarantees that noisy and extreme gradients cannot greatly influence the update steps, and $\hat{v}_t$ would gradually become stable. Similar to Kingma & Ba (2015)'s derivation, in order to

---

**Algorithm 2** AdaX Algorithm

---

1: **Input:** $x \in \mathcal{F}$, step size $\{\alpha_t\}_{t=1}^{T}, \beta_1, \beta_2, \beta_3$
2: **Initialize** $m_0 = 0, v_0 = 0$
3: **for** $t = 1$ **to** $T$ **do**
4:      $g_t = \nabla f_t(x_t)$
5:      $m_t = \beta_1 m_{t-1} + (1 - \beta_3)g_t$
6:      $v_t = (1 + \beta_2)v_{t-1} + \beta_2 g_t^2$
7:      $\hat{v}_t = v_t/[(1 + \beta_2)^t - 1]$ and $V_t = \text{diag}(\hat{v}_t)$
8:      $x_{t+1} = \Pi_{\mathcal{F}, \sqrt{V_t}}(x_t - \alpha_t m_t/\sqrt{\hat{v}_t})$
9: **end for**

---

achieve an unbiased estimate of second moment, we obtain our bias correction term as follows. Let $g_t$ be the gradient at timestep $t$ and further suppose $g_t$'s are drawn from a stationary distribution

$g_t \sim p(g_t)$. Take expectation on both sides of line 6 in Algorithm 2, we get

$$
\begin{aligned}
\mathbb{E}(v_t) &= \mathbb{E}((1 + \beta_2)v_{t-1} + \beta_2 g_t^2) \\
&= \sum_{i=1}^{t} (1 + \beta_2)^{t-i} \beta_2 \mathbb{E}(g_t^2) \\
&= [(1 + \beta_2)^t - 1]\mathbb{E}(g_t^2)
\end{aligned}
\tag{4}
$$

To maintain an accurate second moment, we would naturally divide $v_t$ by $(1 + \beta_2)^t - 1$ in line 7. However, it's worth mentioning that we do not include a first moment correction term $(1 - \beta_1^t)$ as Kingma & Ba (2015) did for the following reason. Consider the Stochastic Gradient Descent with momentum(SGDM) algorithm and Adam's first moment,

$$
\text{SGDM:} \quad m_t = \gamma m_{t-1} + g_t = \sum_{i=1}^{t} \gamma^{t-i} g_i
$$

$$
\text{Adam:} \quad m_t = \beta_1 m_{t-1} + (1 - \beta_1)g_t = (1 - \beta_1)\sum_{i=1}^{t} \beta_1^{t-i} g_i
$$

It can be observed that they have the same form except for the scaling constant $1 - \beta_1$, and therefore the first order bias correction term is counter intuitive. We change the scaling constant from $1 - \beta_1$ to $1 - \beta_3$ in line 5 in our algorithm to obtain a more general form of the moment expression, and when $\beta_3 \neq \beta_1$, it helps to scale the hyper-parameters (such as step size, weight decay) of adaptive algorithms to the same size as SGD. Next we show that our algorithm ensures the positive semi-definiteness of $\Gamma_t$ and hence does not have the non-convergence issue of Adam. Consider the following lemma which leads to the conclusion that $\Gamma_t$ is positive semi-definite in our algortihm,

**Lemma 4.1** *Algorithm 2 ensures that the matrix $\frac{V_t}{\alpha_t^2} - \frac{V_{t-1}}{\alpha_{t-1}^2} \succeq 0$*

Finally, we provide the convergence analysis of our algorithm in both convex and non-convex settings. Using the analysis framework by Zinkevich (2003), the following theorem states that we are able to obtain a regret bound of $\mathcal{O}(\sqrt{T})$, which is the same as the results of Reddi et al. (2018). A domain $\mathcal{F}$ is said to have bounded diameter if $\|x - y\|_\infty \leq D_\infty, \forall x, y \in \mathcal{F}$ for some $D_\infty \in \mathbb{R}$.

**Theorem 4.1** *Let $\{x_t\}$ and $\{v_t\}$ be the sequences obtained from Algorithm 2, $\alpha_t = \alpha/\sqrt{t}, \beta_{1,1} = \beta_1, \beta_{1,t} \leq \beta_1,$ for all $t \in [T]$ and $\beta_{2t} = \beta_2/t, \beta_{3t} = 1 - 1/\sqrt{t}$. Assume that $\mathcal{F}$ has bounded diameter $D_\infty$ and $\|\nabla f_t(x)\| \leq G_\infty$ for all $t \in [T]$ and $x \in \mathcal{F}$. Then for $x_t$ generated using Algorithm 2, we have the following bound on the regret.*

$$
R_T \leq \frac{D_\infty^2}{2\alpha_T(1 - \beta_1)}\sum_{i=1}^{d} \hat{v}_{T,i}^{1/2} + \frac{D_\infty^2}{2(1 - \beta_1)}\sum_{t=1}^{T}\sum_{i=1}^{d} \frac{\beta_{1t}\hat{v}_{t,i}^{1/2}}{\alpha_t} + \frac{\alpha C\sqrt{1 + \log T}}{(1 - \beta_1)^3\sqrt{\beta_2}}\sum_{i=1}^{d} \|g_{1:T,i}\|_2
\tag{5}
$$

The following corollary follows naturally from the above theorem.

**Corollary 4.1** *Suppose $\beta_{1t} = \beta_1\lambda^{t-1}$ in Theorem 4.1, then we have*

$$
R_T \leq \frac{D_\infty^2\sqrt{T}}{2\alpha(1 - \beta_1)}\sum_{i=1}^{d} \hat{v}_{T,i}^{1/2} + \frac{d\beta_1 D_\infty^2 G_\infty}{2\alpha(1 - \beta_1)(1 - \lambda)^2} + \frac{\alpha C\sqrt{1 + \log T}}{(1 - \beta_1)^3\sqrt{\beta_2}}\sum_{i=1}^{d} \|g_{1:T,i}\|_2
\tag{6}
$$

Analyzing optimization algorithms in a non-convex setting is slightly different from the convex case, where instead of the average regret, stationarity in gradient is utilized to show convergence in time. Following Chen et al. (2019), suppose we are minimizing a cost function $f$ that satisfies the following three assumptions

**A1.** $f$ is differentiable and has $L$-Lipschitz gradient, i.e. $\forall x, y, \|\nabla f(x) - \nabla f(y)\| \leq L\|x - y\|$. and $f(x^*) > \infty$ where $x^*$ is the optimal solution.

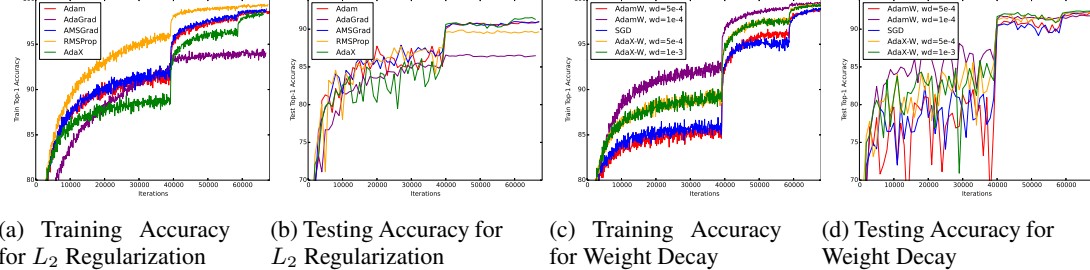

(a) Training Accuracy for $L_2$ Regularization

(b) Testing Accuracy for $L_2$ Regularization

(c) Training Accuracy for Weight Decay

(d) Testing Accuracy for Weight Decay

Figure 1: Training and Testing Accuracy on CIFAR-10

| Method | Top 1 Acc | Method | Top 1 Acc |
|---|---|---|---|
| Adam | $90.96 \pm 0.03$ | AdamW, wd=1e-4 | $91.86 \pm 0.04$ |
| AdaGrad | $86.56 \pm 0.07$ | AdamW, wd=5e-4 | $92.12 \pm 0.05$ |
| AMSGrad | $90.98 \pm 0.04$ | SGDM | $92.30 \pm 0.09$ |
| RMSProp | $89.64 \pm 0.05$ | **AdaX-W, wd=5e-4(ours)** | **$92.32 \pm 0.04$** |
| **AdaX(ours)** | **$91.60 \pm 0.10$** | **AdaX-W, wd=1e-3(ours)** | **$92.51 \pm 0.07$** |

Table 2: Validation accuracy on CIFAR-10.

**A2**. Suppose that the true gradients and the noisy gradients are bounded i.e. $\|\nabla f(x_t)\| \le G_\infty, \|g_t\| \le G_\infty, \forall t \ge 1$. Also, $\|\alpha_t \frac{m_t}{\sqrt{v_t}}\| \le G$ for some $G > 0$

**A3**. The noisy gradient is unbiased and the noise is independent, i.e. $g_t = \nabla f(x_t) + \eta_t, \mathbb{E}[\eta_t] = 0$ and $\eta_i$ is independent of $\eta_j$ if $i \ne j$.

then we would obtain the following theorem and corollary, which prove that AdaX converges with a speed close to AMSGrad as mentioned by Chen et al. (2019).

**Theorem 4.2** *Let $\{x_t\}$ and $\{v_t\}$ be the sequences obtained from Algorithm 2, $\alpha_t = \alpha/\sqrt{t}, \beta_{1,1} = \beta_1, \beta_{1,t} \le \beta_1$, for all $t \in [T]$ and $\beta_{2t} = \beta_2/t, \beta_{3t} = \beta_1$. Assume that $\|\nabla f_t(x)\| \le G_\infty$ for all $t \in [T]$ and $x \in \mathcal{F}$. Then for $x_t$ generated using Algorithm 2, we have the following bound.*

$$\min_{t \in [T]} \mathbb{E}\left[\|\nabla f(x_t)\|^2\right] \le \frac{G_\infty}{\alpha \log(1+T)}\left(\frac{C_1 G_\infty^2 \alpha^2}{c^2} + \frac{C_2 d\alpha}{c} + \frac{C_3 d\alpha^2}{c^2} + C_4\right) \tag{7}$$

*where $C_1, C_2, C_3, C_4$ are constants independent of $T$*

**Corollary 4.2** *Suppose $\beta_{2t} = \beta_2$, with the other assumptions same as in Theorem 4.2, we have*

$$\min_{t \in [T]} \mathbb{E}\left[\|\nabla f(x_t)\|^2\right] \le \frac{G_\infty}{\alpha\sqrt{T}}\left(\frac{C_1 G_\infty^2 \alpha^2}{c^2} + \frac{C_2 d\alpha}{c} + \frac{C_3 d\alpha^2}{c^2} + C_4\right) \tag{8}$$

## 5 EXPERIMENTS

In this section, we evaluate the performance of AdaX on various tasks in comparison with SGD with Nesterov momentum (SGDM), Adam(W), and many other common optimizers. The implementation of AdaX consists of two parts, AdaX and AdaX-W, representing using $L_2$ regularization and standard weight decay in the algorithm respectively as discussed in Loshchilov & Hutter (2019). We relegate the detailed implementation of AdaX to the Appendix. We show that AdaX, combined with a proper weight decay, is capable of performing better than Adam and SGDM in many tasks.

### 5.1 CONVOLUTIONAL NEURAL NETWORK ON CIFAR-10

Using ResNet-20 proposed by He et al. (2016), we verified the performance of AdaX on CIFAR-10 (Krizhevsky et al., 2009) image classification task. In our experiments, we utilized a learning rate

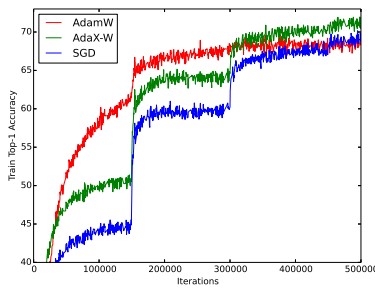
(a) Training Top-1 Accuracy on ImageNet

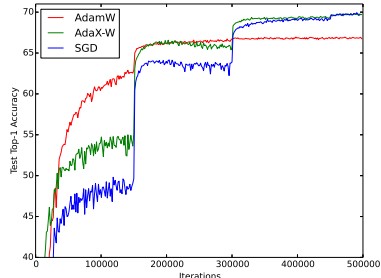
(b) Testing Top-1 Accuracy on ImageNet

Figure 2: Training and Testing Results on ImageNet.

(a) Validation accuracy on ImageNet and IoU on VOC2012 Segmentation

(b) Validation perplexity on One Billion Word for language modeling.

| Method | ImageNet Top 1 | Method | VOC2012 IoU | Method | Validation PPL |
|---|---|---|---|---|---|
| SGDM | **69.90** | SGDM | 76.28 | Adam | 36.90 |
| AdamW | 66.92 | AdamW | 74.62 | **AdaX(ours)** | **35.22** |
| **AdaX-W(ours)** | **69.87** | RMSProp | 58.70 | | |
| | | AMSGrad | 73.62 | | |
| | | AdaX-W | **76.53** | | |

Table 3: Performance of AdaX on ImageNet, VOC2012 Segmentation and One Billion Word

schedule that the initial step size was scaled down by 0.1 and 0.01 at the 100-th and the 150-th epoch. The experimental results in Table 2 corresponded to our theoretical finding that Adam was actually taking steps that were "too large" and would potentially converge to local minimum at the end.

$L_2$ **Regularization.** We used $L_2$ regularization of strength 5e-4 for all the optimizers. As shown in Figure 1a and 1b, although AdaX was relatively slow at the beginning compared with other optimizers, its testing accuracy quickly caught up with the others after the first learning rate decrease and became the highest (91.6) at last.

**Weight Decay.** The baseline was trained with SGDM with weight decay 5e-4. Adam with weight decay, named AdamW (Loshchilov & Hutter, 2019) was also trained for comparisons. Although AdamW with 1e-4 weight decay converged much faster than SGDM and AdaX-W, its final accuracy could not catch up with the other two (see Figure 1c & 1d).A higher weight decay could potentially help AdamW achieve a better performance (92.1), but it was as slow as SGDM. On the other hand, AdaX-W with step size rate 0.5 and 5e-4 weight decay converged fast and yielded the same performance as SGDM(92.32). The best result we obtained was AdaX-W with step size 0.25 and 1e-3 weight decay, which resulted in 92.51 Top-1 accuracy, even slightly higher than SGDM.

## 5.2 CONVOLUTIONAL NEURAL NETWORK ON IMAGENET

We also conducted experiments to examine the performance of AdaX-W on ImageNet (Deng et al., 2009). SGDM, AdaX-W, and AdamW were used to train a ResNet-18 model on ImageNet, with a standard 1e-4 weight decay and 0.1, 0.5, 1e-3 step sizes as in CIFAR-10 respectively. A warm up scheme was applied in the initial 25k iterations (Goyal et al., 2017), and then the step size was multiplied by 0.1 at the 150k, 300k and 450k-*th* iteration steps. As observed from Figure 2, although AdamW was fast at the beginning of training, its test accuracy stagnated after the second learning rate decrease. AdaX-W, on the other hand, converged faster than SGDM without loss of testing accuracy (69.87), as shown in Table 3a.

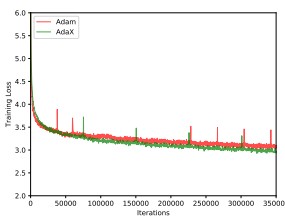 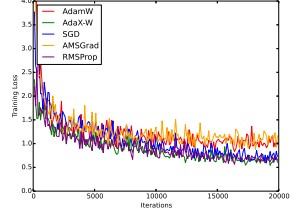 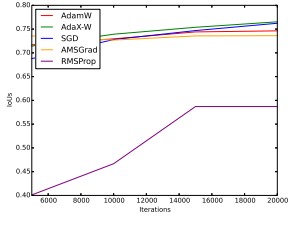

(a) Training Dynamics on One Billion Word.

(b) Training Loss on VOC2012 Segmentation

(c) Testing IoU

Figure 3: (a) Traning loss curves for Adam and AdaX on One Billion Word. (b, c) Training Loss and Testing Results on VOC2012 Segmentation task. In (c), dashed lines are mean accuracy values and solid lines are Intersection over Union (IoU) values

## 5.3 RECURRENT NEURAL NETWORK ON LANGUAGE MODELING

AdaX has also been validated on One Billion Word (Chelba et al., 2013) dataset of language modeling task. For the One Billion Word, we used a two-layer LSTMs with 2048 hidden states and sampled softmax. The global experiment settings in the released public code Rdspring1 was adopted in this study. For both vanilla Adam and AdaX, the LSTMs were trained using for 5 epochs, with learning rate decaying to 1e-8 linearly. Similarly, the weight decay for AdaX was set to 0. Note that the regular SGD is not suitable in this task, so it is not included in the comparison.

The training loss and the validation perplexity is shown in Figure 3a and Table 3b. We can see that the AdaX outperforms the Adam baseline by a significant margin (35.22 vs. 36.90). Moreover, similar to the effect on image classification tasks described above, AdaX starts a little slower at the early stage, but it soon surpasses Adam on both training and validation performance.

## 5.4 TRANSFER LEARNING

To further examine the robustness of AdaX in transfer learnings such as semantic segmentation, we evaluated its performance on the PASCAL VOC2012 augmented dataset (Everingham et al., 2014) (Hariharan et al., 2011). The classic Deeplab-ASPP model (Chen et al., 2016) was adopted with a ResNet-101 backbone pretrained on MS-COCO dataset(Lin et al., 2014). Adaptive methods are seldom used in semantic segmentation tasks because of their bad performances, but AdaX can surprisingly be applied to train these models as well. The initial step sizes for different algortithms was recorded in section A.7 and all other parameters stuck to the default setting in Chen et al. (2016). We evaluated the algorithms' performances at the 5k, 10k, 15k and 20k iterations using intersection over union (IoU). As can be observed in Figure 3c, AdaX-W trained faster than SGDM and obtained a higher IoU (76.5) at the same time. On the other hand, AdamW was not capable of obtaining comparable results.

## 5.5 COMPARISON OF SECOND MOMENT DESIGN

Finally, we compared the second moment design of Adam and AdaX empirically and showed the instability of Adam's second moment could have a large impact on its performance while our design was stable and robust. We first evaluated the performance of different algorithms in our synthetic example (3). The problem parameters were set to be $C = 1e-3, \lambda = 0.9999, x_0 = 1$. To ensure fair comparisons, default hyperparameters were chosen for all the algorithms, specifically $\alpha_0 = 0.1, \gamma = 0.9$ for SGDM, $\alpha_0 = 1e-3, \beta_1 = 0.9, \beta_2 = 0.999$ for Adam, and $\alpha_0 = 0.15, \beta_1 = 0.9, \beta_2 = 1e-4, \beta_3 = 0.999$ for AdaX. As shown in Figure 4a and 4b, SGD and AdaX quickly converged under the strong gradient decrease information and could potentially reach the global minimum if initialized well. However, the update steps of Adam decreased with a much slower rate, which resulted in substantial changes in $x$ and ultimately lead the algorithm to the local minimum.

Zaheer et al. (2018) found that Adam's performance could be affected by different values of $\epsilon$, which was originally designed to avoid zeros in the denominator. Such a finding revealed the instability

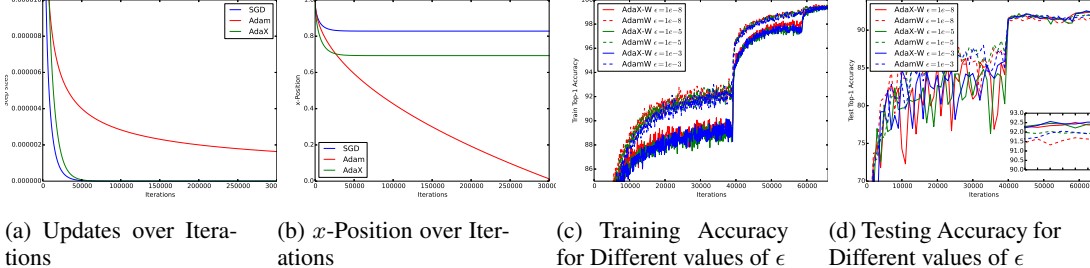

(a) Updates over Iterations

(b) $x$-Position over Iterations

(c) Training Accuracy for Different values of $\epsilon$

(d) Testing Accuracy for Different values of $\epsilon$

Figure 4: Second momentum stability comparisons. (a),(b). Training updates and x-positions on Problem (3). (c),(d) Training and testing results of AdamW and AdaX-W with different $\epsilon$'s

of Adam's second moment. We verified their claim by testing AdamW with different values of $\epsilon \in \{1e-8, 1e-5, 1e-3\}$ on the CIFAR-10 dataset. We found that larger values of $\epsilon$, since it helped to stabilize very small second moment, did improve AdamW's performance by around 0.35 percent accuracy on CIFAR-10. However, AdaX-W's performance was not affected by different choices of $\epsilon$ as shown in Figure 4d, which again proved our claim that a long term memory design was more stable than the exponential moving average in Adam.

The experiments shown above verify the effectiveness of AdaX, showing that the accumulated long-term past gradient information can enhance the model performance, by getting rid of the second moment instability in vanilla Adam. It is also worth noticing that the computational cost for each step of AdaX and Adam are approximately the same, as they both memorize the first and second momentum in the past. Therefore AdaX enables one to get higher performance than Adam in various tasks using the same training budget.

## 6 CONCLUSION

In this paper, we present a novel optimization algorithm named AdaX in order to improve the performance of traditional adaptive methods. We first extend the non-convergence issue of Adam to a non-convex case, and show that Adam's fast convergence impairs its convergence. We then propose our variant of Adam, analyze its convergence, and evaluate its performance on various learning tasks. Our theoretical analysis and experimental results both show that AdaX is more stable and performs better than Adam in various tasks. In the future, more experiments still need to be performed to evaluate the overall performance of AdaX and AdaX-W. Moreover, our paper is a first step into designing adaptive learning rates in ways different from simple and exponential average methods. Other new and interesting designs should also be examined. We believe that new adaptive algorithms that outperform AdaX in convergence rate and performance still exist and remain to explore.

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

# A  APPENDIX

## A.1  PROOFS OF LEMMA 3.1, LEMMA 3.2

We consider a one dimensional non-convex case where $\{f_t\}$ are a sequence of linear functions that have decreasing gradients in the long term. We want to show that because Adam trusts its current gradient as the second moment, its step sizes are too large and the algorithm would converge to a suboptimal solution. Let constant $C$ be the initial gradient, define cost function $f_t$ as follows:

$$f_t(x) = \begin{cases} C\lambda^{t-1}x, & \text{for } x \geq 0 \\ \dfrac{C^2}{1-\lambda}, & \text{for } x < 0 \end{cases} \qquad \forall t \geq 1 \tag{9}$$

where $\lambda$ is the decreasing factor of gradient. Consider $\mathcal{F} = [-2, C/(1-\lambda)]$, then it's obvious that the minimum regret is obtained at $x = 0$. Let the initial step size be $\alpha_1 = \alpha$, we then consider the performances of different algorithms in this setting.

(**SGD**). We first show that without the momentum, vanilla SGD is able to converge to the optimum solution and avoid the trap. Take derivative with respect to $x$, we obtain that

$$\nabla f_t(x) = C\lambda^{t-1}, \text{ for } x \geq 0$$
$$\sum_{t=1}^{\infty} \nabla f_t(x) = \sum_{t=1}^{\infty} C\lambda^{t-1} = \frac{C}{1-\lambda} \tag{10}$$

Therefore, even if we set $\alpha_t = \alpha, \forall t \geq 1$, as long as the initial point $x_0 \geq \frac{\alpha C}{1-\lambda}$, SGD is able to avoid the trap. If $\alpha_t = \alpha/\sqrt{t}$, then the condition can be even less strict: $x_0 \geq \sum_{t=1}^{\infty} \frac{\alpha C\lambda^{t-1}}{\sqrt{t}}$. SGD is able to converge to the optimum if the equal signs are true.

(**Adam**). We consider the Adam algorithm with the following parameter setting:

$$\beta_1 = 0, 0 < \sqrt{\beta_2} < \lambda < 1, \text{ and } \alpha_t = \frac{\alpha}{\sqrt{t}} \tag{11}$$

Note that this parameter setting of Adam is the same as RMSProp, but we can further show that even if $\beta_1 \neq 0$, we still obtain similar results. Consider how $v_t$ changes in time, before it reaches the negative region, the gradients are positive and

$$\begin{aligned} v_t &= \beta_2 v_{t-1} + (1-\beta_2)(C\lambda^{t-1})^2 \\ &= \sum_{i=1}^{t} \beta_2^{t-i}(1-\beta_2)(C\lambda^{i-1})^2 \\ &= \frac{(1-\beta_2)C^2}{\lambda^2}\beta_2^t \sum_{i=1}^{t}(\frac{\lambda^2}{\beta_2})^i \\ &= \frac{(1-\beta_2)C^2(\lambda^{2t}-\beta_2^t)}{\lambda^2 - \beta_2} \end{aligned} \tag{12}$$

Note that $\lambda > \sqrt{\beta_2}$, therefore the update rule is:

$$\begin{aligned} x_{t+1} &= x_t - \alpha_t \frac{g_t}{\sqrt{v_t}} = x_t - \alpha_t \frac{\sqrt{\lambda^2 - \beta_2}\lambda^{t-1}}{\sqrt{(1-\beta_2)(\lambda^{2t}-\beta_2^t)}} = x_t - \alpha_t \frac{\sqrt{\lambda^2 - \beta_2}\lambda^{-1}}{\sqrt{(1-\beta_2)(1-(\frac{\beta_2}{\lambda^2})^t)}} \\ &\leq x_t - \frac{\alpha}{\sqrt{t}}\sqrt{\frac{\lambda^2 - \beta_2}{\lambda^2(1-\beta_2)}} \end{aligned} \tag{13}$$

Note that the series $\sum_{t=1}^{\infty} \frac{1}{\sqrt{t}}$ diverges, hence Adam would always reach the negative region. Same argument applies as long as $\alpha_t \geq \alpha/t$. We would emphasize here that the bias correction term in Adam does not change the final result as $1 - \beta_2^t \geq 1 - \beta_2$ and therefore the update steps are still bounded. We could further show that when $\beta_1 \neq 0, \beta_1 < \sqrt{\beta_2}$, Adam will still go to the negative region. Since $m_t = \beta_1 m_{t-1} + (1 - \beta_1)g_t = \sum_{i=1}^{t}(1 - \beta_1)\beta_1^{t-i}g_i$, therefore

$$
\begin{aligned}
\frac{m_t}{\sqrt{v_t}} &= \frac{(1 - \beta_1)(\lambda^t - \beta_1^t)}{\lambda - \beta_1} \cdot \frac{\sqrt{\lambda^2 - \beta_2}}{\sqrt{(1 - \beta_2)(\lambda^{2t} - \beta_2^t)}} \\
&= \frac{(1 - \beta_1)\sqrt{\lambda^2 - \beta_2}}{(\lambda - \beta_1)\sqrt{(1 - \beta_2)}} \cdot \frac{\lambda^t - \beta_1^t}{\sqrt{\lambda^{2t} - \beta_2^t}} \\
&= \frac{(1 - \beta_1)\sqrt{\lambda^2 - \beta_2}}{(\lambda - \beta_1)\sqrt{1 - \beta_2}} \cdot \frac{1 - (\beta_1/\lambda)^t}{\sqrt{1 - (\beta_2/\lambda^2)^t}} \\
&\geq \frac{(1 - \beta_1)\sqrt{\lambda^2 - \beta_2}}{(\lambda - \beta_1)\sqrt{1 - \beta_2}} \cdot (1 - \frac{\beta_1}{\lambda})
\end{aligned}
\tag{14}
$$

Since the update steps are lower bounded, the algorithm would still go to the negative region.

**(AMSGrad)**. We now evaluate the performance of AMSGrad in our formulated problem. Note that $v_t$ in AMSGrad would take the same form as Adam, and $\hat{v}_t = \max\{v_t\}_{i=1}^{t}$. We suppose that the maximum is obtained at $v_1$ as an example, then

$$
\begin{aligned}
\hat{v}_t = v_1 &= \frac{(1 - \beta_2)C^2(\lambda^2 - \beta_2)}{\lambda^2 - \beta_2} = (1 - \beta_2)C^2 \\
\frac{g_t}{\sqrt{\hat{v}_t}} &= \frac{g_t}{\sqrt{v_1}} = \frac{\lambda^{t-1}}{\sqrt{1 - \beta_2}}
\end{aligned}
\tag{15}
$$

As we can see, AMSGrad partially solves the problem of Adam and restores the gradient decrease information as its $v_t$ is lower bounded. If the maximum of $v_t$ is obtained before the parameters enter the trap, AMSGrad could possibly have a better performance in this problem as it prevents the update steps from being too large. However, one important determining factor is the time when the maximum value is obtained. If $v_t$ in fact keeps increasing before a very large number $T$, then AMSGrad would have the same performance as Adam. We explain the above intuition as follows. Let $h(t) = \lambda^{2t} - \beta_2^t$, then

$$
\frac{dh(t)}{dt} = \ln \lambda^2 \cdot \lambda^{2t} - \ln \beta_2 \cdot \beta_2^t
\tag{16}
$$

If $\frac{dh(t)}{dt} \geq 0$, we have

$$
\ln \lambda^2 \cdot \lambda^{2t} - \ln \beta_2 \cdot \beta_2^t \geq 0
$$
$$
(\frac{\lambda^2}{\beta_2})^t \leq \frac{\ln \beta_2}{\ln \lambda^2}
$$

When the equal sign is true, we have

$$
\begin{aligned}
t &= \log_{\frac{\lambda^2}{\beta_2}} \frac{\ln \beta_2}{\ln \lambda^2} = \frac{\ln \frac{\ln \beta_2}{\ln \lambda^2}}{\ln \frac{\lambda^2}{\beta_2}} \\
\lim_{\beta_2 \to \lambda^2} t &= \lim_{\beta_2 \to \lambda^2} \frac{\frac{1}{\beta_2 \ln(\beta_2)}}{-\frac{1}{\beta_2}} = -\frac{1}{\ln(\lambda^2)} \\
\lim_{\lambda \to 1^-} t &= \infty
\end{aligned}
\tag{17}
$$

The first equal sign in the first limit is due to L'Hospital's rule. Therefore, the value of $T$ where $v_T = \max\{v_t\}$ depends on the difference between $\beta_2$ and $\lambda^2$, and the value of $\lambda$. If $\beta_2$ is close to $\lambda^2$ or $\lambda$ is close to 1, then $v_t$ needs a large number of steps to obtain the maximum. In such cases, AMSGrad may not able to help Adam. Specifically, for a fixed $\beta_2$, since $\lim_{\lambda \to 1^-} t = \infty$. and

$$g_t/\sqrt{v_t} = \sqrt{\frac{\lambda^2 - \beta_2}{\lambda^2(1 - \beta_2)}} = \frac{1}{\sqrt{1 - \beta_2}}\sqrt{1 - \frac{\beta_2}{\lambda^2}} \tag{18}$$

we know that larger $\lambda^2$ will lead to both larger update steps and larger $T$ when the maximum is obtained, hence $\exists \lambda \in (0, 1)$, such that AMSGrad cannot avoid the trap.

**(AdaX).** We provide the performance of AdaX in this problem for completeness. We only show for the case when $\beta_1 = \beta_3 = 0$, but the same results hold when the first order momentum is used.

$$
\begin{aligned}
\hat{v}_t &= \frac{1}{(1 + \beta_2)^t - 1}\left[\sum_{i=1}^{t}\beta_2(1 + \beta_2)^{t-i}(C\lambda^{i-1})^2\right] \\
&= \frac{(1 + \beta_2)^t\beta_2 C^2}{(1 + \beta_2)^t - 1}\lambda^{-2}\sum_{i=1}^{t}(\frac{\lambda^2}{1 + \beta_2})^i \\
&= \frac{(1 + \beta_2)^t\beta_2 C^2}{(1 + \beta_2)^t - 1}\cdot\frac{1 - (\frac{\lambda^2}{1+\beta_2})^t}{1 + \beta_2 - \lambda^2} \\
&= \frac{\beta_2 C^2}{1 + \beta_2 - \lambda^2}\cdot\frac{(1 + \beta_2)^t - \lambda^{2t}}{(1 + \beta_2)^t - 1}
\end{aligned}
\tag{19}
$$

$$\frac{g_t}{\sqrt{\hat{v}_t}} = \sqrt{\frac{1 + \beta_2 - \lambda^2}{\beta_2}}\cdot\frac{\lambda^{t-1}\sqrt{(1 + \beta_2)^t - 1}}{\sqrt{(1 + \beta_2)^t - \lambda^{2t}}} \leq \sqrt{\frac{1 + \beta_2 - \lambda^2}{\beta_2}}\lambda^{t-1} \tag{20}$$

As we can see, AdaX successfully restores the gradient decrease information and controls the decrease speed by an almost fixed parameter instead of trusting the gradient at a certain moment, and is therefore expected to perform better than AMSGrad since its step sizes are not affected by extreme gradients. With a suitable initial step size and starting point, AdaX is able to avoid the designed trap.

### A.2    PROOF OF LEMMA 4.1

*Proof.*

$$
\begin{aligned}
\frac{V_t}{\alpha_t^2} &= \frac{t}{\alpha^2}\frac{\sum_{i=1}^{t}(1 + \beta_2)^{t-i}\beta_2 g_i^2}{(1 + \beta_2)^t - 1} \\
&\succeq \frac{t - 1}{\alpha^2}\frac{\sum_{i=1}^{t}(1 + \beta_2)^{t-i}\beta_2 g_i^2}{(1 + \beta_2)^t - (1 + \beta_2)} \\
&\succeq \frac{t - 1}{\alpha^2}\frac{\sum_{i=1}^{t}(1 + \beta_2)^{t-i}\beta_2 g_i^2 - \beta_2 g_t^2}{(1 + \beta_2)^t - (1 + \beta_2)} \\
&= \frac{t - 1}{\alpha^2}\frac{\sum_{i=1}^{t-1}(1 + \beta_2)^{t-1-i}\beta_2 g_i^2}{(1 + \beta_2)^{t-1} - 1} = \frac{V_{t-1}}{\alpha_{t-1}^2}
\end{aligned}
\tag{21}
$$

where in the first inequality we utilize the fact that $(1 + \beta_2)^t \geq 1 + t\beta_2$ and hence $\frac{t}{(1+\beta_2)^t - 1} \geq \frac{t-1}{(1+\beta_2)^t - (1+\beta_2)}$. Intuitively, it is easier to see this inequality if we simply let $\beta_2$ to be a small number such as 1e-4 in our implementation, then the denominator doesn't change much while the numerator decreases.

### A.3    AUXILLARY LEMMAS FOR CONVERGENCE ANALYSIS

**Lemma A.1** *Assume that $\beta_{21} = \beta_2, \beta_{2t} = \beta_2/t$, with $\beta_2 \in (0, 1)$ and $\hat{v}_t = [(1 + \beta_{2t})v_{t-1} + \beta_{2t}g_t^2]/[(1 + \beta_{2t})^t - 1], V_t = diag(\hat{v}_t)$, then we have $\frac{V_t}{\alpha_t^2} \succeq \frac{V_{t-1}}{\alpha_{t-1}^2}$*

*Proof:* Similar to **Lemma 4.1** in the algorithm section, we have

$$
\begin{aligned}
\frac{V_t}{\alpha_t^2} &= \frac{t}{\alpha^2} \frac{\sum_{i=1}^{t} \beta_{2i} \Pi_{k=1}^{t-i}(1 + \beta_{2(t-k+1)}) g_i^2}{(1 + \beta_{2t})^t - 1} \\
&= \frac{t}{\alpha^2} \frac{\sum_{i=1}^{t} \frac{\beta_2}{i} \Pi_{k=1}^{t-i}(1 + \frac{\beta_2}{t-k+1}) g_i^2}{(1 + \frac{\beta_2}{t})^t - 1} \\
&\succeq \frac{t}{\alpha^2} \frac{\sum_{i=1}^{t-1} \frac{\beta_2}{i} \Pi_{k=1}^{t-i}(1 + \frac{\beta_2}{t-k+1}) g_i^2}{(1 + \frac{\beta_2}{t})^t - 1} \\
&= \frac{t}{\alpha^2} \frac{\sum_{i=1}^{t-1} \frac{\beta_2}{i} \Pi_{k=1}^{t-1-i}(1 + \frac{\beta_2}{t-k+1}) g_i^2}{(1 + \frac{\beta_2}{t})^{t-1} - (1 + \frac{\beta_2}{t})^{-1}} \\
&\succeq \frac{t-1}{\alpha^2} \frac{\sum_{i=1}^{t-1} \frac{\beta_2}{i} \Pi_{k=1}^{t-1-i}(1 + \frac{\beta_2}{t-k+1}) g_i^2}{(1 + \frac{\beta_2}{t-1})^{t-1} - 1} = \frac{V_{t-1}}{\alpha_{t-1}^2}
\end{aligned}
\tag{22}
$$

The first inequality comes from deleting the last term $\frac{\beta_2}{t} g_t^2$ and second one comes from the following fact:

$$
\begin{aligned}
(1 + \frac{\beta_2}{t})^{t-1} - (1 + \frac{\beta_2}{t})^{-1} &= (1 + \frac{\beta_2}{t})^{t-1} - \frac{t}{t + \beta_2} \\
&\leq 1 + \frac{t-1}{t} \beta_2 + \binom{t-1}{2}(\frac{\beta_2}{t})^2 + \cdots + \binom{t-1}{t-1}(\frac{\beta_2}{t})^{t-1} - 1 + \frac{\beta_2}{t + \beta_2} \\
&= 1 + \beta_2 + \binom{t-1}{2}(\frac{\beta_2}{t})^2 + \cdots + \binom{t-1}{t-1}(\frac{\beta_2}{t})^{t-1} - 1 + (\frac{\beta_2}{t + \beta_2} - \frac{\beta_2}{t}) \\
&\leq 1 + \beta_2 + \binom{t-1}{2}(\frac{\beta_2}{t-1})^2 + \cdots + \binom{t-1}{t-1}(\frac{\beta_2}{t-1})^{t-1} - 1 \\
&= (1 + \frac{\beta_2}{t-1})^{t-1} - 1
\end{aligned}
\tag{23}
$$

Therefore the positive semi-definiteness is satisfied.

**Lemma A.2** *For the parameter settings and conditions assumed in **Theorem 4.1**, we have*

$$
\sum_{t=1}^{T} \frac{\beta_{1t} \alpha_t}{2(1 - \beta_{1t})} \| V_t^{-1/4} m_{t-1} \|^2 \leq \frac{\alpha C \sqrt{1 + \log T}}{(1 - \beta_1)^2 \sqrt{\beta_2}} \sum_{i=1}^{d} \| g_{1:T,i} \|_2
\tag{24}
$$

*Proof:* We first analyze with the following process directly from the update rules, note that

$$
\begin{aligned}
m_{T,i} &= \sum_{j=1}^{T} (1 - \beta_{3j}) \Pi_{k=1}^{T-j} \beta_{1(T-k+1)} g_{j,i} \\
\hat{v}_{T,i} &= \frac{1}{(1 + \beta_{2T})^T - 1} \sum_{j=1}^{T} \beta_{2j} \Pi_{k=1}^{T-j}(1 + \beta_{2(T-k+1)}) g_{j,i}^2
\end{aligned}
\tag{25}
$$

$$\sum_{t=1}^{T} \alpha_t \|V_t^{-1/4} m_{t-1}\|^2$$

$$= \sum_{t=1}^{T-1} \alpha_t (1 - \beta_{1t}) \|V_t^{-1/4} m_{t-1}\|^2 + \alpha_T \sum_{i=1}^{d} \frac{m_{T,i}^2}{\sqrt{\hat{v}_{T,i}}}$$

$$= \sum_{t=1}^{T-1} \alpha_t (1 - \beta_{1t}) \|V_t^{-1/4} m_{t-1}\|^2 + \alpha \sqrt{(1 + \beta_{2T})^T - 1} \sum_{i=1}^{d} \frac{(\sum_{j=1}^{T} (1 - \beta_{3j}) \Pi_{k=1}^{T-j} \beta_{1(T-k+1)} g_{j,i})^2}{\sqrt{T \sum_{j=1}^{T} \beta_{2j} \Pi_{k=1}^{T-j} (1 + \beta_{2(T-k+1)}) g_j^2}}$$

$$\leq \sum_{t=1}^{T-1} \alpha_t (1 - \beta_{1t}) \|V_t^{-1/4} m_{t-1}\|^2 + \alpha \sqrt{(1 + \beta_{2T})^T - 1}$$

$$c \sum_{i=1}^{d} \frac{(\sum_{j=1}^{T} \Pi_{k=1}^{T-j} \beta_{1(T-k+1)})(\sum_{j=1}^{T} (1 - \beta_{3j})^2 \Pi_{k=1}^{T-j} \beta_{1(T-k+1)} g_{j,i}^2)}{\sqrt{T \sum_{j=1}^{T} \beta_{2j} \Pi_{k=1}^{T-j} (1 + \beta_{2(T-k+1)}) g_{j,i}^2}}$$

$$\leq \sum_{t=1}^{T-1} \alpha_t (1 - \beta_{1t}) \|V_t^{-1/4} m_{t-1}\|^2 + \alpha \sqrt{(1 + \beta_{2T})^T - 1} \sum_{i=1}^{d} \frac{(\sum_{j=1}^{T} \beta_1^{T-j})(\sum_{j=1}^{T} (1 - \beta_{3j})^2 \beta_1^{T-j} g_{j,i}^2)}{\sqrt{T \sum_{j=1}^{T} \beta_{2j} \Pi_{k=1}^{T-j} (1 + \beta_{2(T-k+1)}) g_j^2}}$$

$$\leq \sum_{t=1}^{T-1} \alpha_t (1 - \beta_{1t}) \|V_t^{-1/4} m_{t-1}\|^2 + \frac{\alpha \sqrt{(1 + \beta_{2T})^T - 1}}{(1 - \beta_1) \sqrt{\beta_2 T}} \sum_{i=1}^{d} \frac{\sum_{j=1}^{T} (1 - \beta_{3j}) \beta_1^{T-j} g_{j,i}^2}{\sqrt{\sum_{j=1}^{T} \frac{1}{j} \Pi_{k=1}^{T-j} (1 + \beta_{2(T-k+1)}) g_{j,i}^2}}$$

$$\leq \sum_{t=1}^{T-1} \alpha_t (1 - \beta_{1t}) \|V_t^{-1/4} m_{t-1}\|^2 + \frac{\alpha \sqrt{(1 + \beta_{2T})^T - 1}}{(1 - \beta_1) \sqrt{\beta_2 T}} \sum_{i=1}^{d} \sum_{j=1}^{T} \frac{\frac{1}{\sqrt{j}} \beta_1^{T-j} g_{j,i}^2}{\sqrt{\frac{1}{j} \Pi_{k=1}^{T-j} (1 + \beta_{2(T-k+1)}) g_{j,i}^2}}$$

$$\leq \sum_{t=1}^{T-1} \alpha_t (1 - \beta_{1t}) \|V_t^{-1/4} m_{t-1}\|^2 + \frac{\alpha C}{(1 - \beta_1) \sqrt{\beta_2 T}} \sum_{i=1}^{d} \sum_{j=1}^{T} \beta_1^{T-j} |g_{j,i}|$$

$$\tag{26}$$

where the first inequality is due to an application of Cauchy-Schwarz inequality. The second inequality is due to the fact that $\beta_{1t} \leq \beta_1, \forall t$. The third inequality follows from $\sum_{j=1}^{T} \beta_1^{T-j} \leq 1/(1 - \beta_1)$ and the fact that $1 - \beta_{3j} \leq 1$. The fourth one comes from only keeping one of the positive terms in the denominator. The final one is from the fact that $\sqrt{(1 + \beta_{2T})^T - 1} \leq C$ for a constant $C = \sqrt{e^{\beta_2} - 1}$. By using induction on all the terms in the equation, we are able to further bound it.

$$\sum_{t=1}^{T} \alpha_t \|V_t^{-1/4} m_{t-1}\|^2 \leq \sum_{t=1}^{T} \frac{\alpha C}{(1 - \beta_1) \sqrt{\beta_2 t}} \sum_{i=1}^{d} \sum_{j=1}^{t} \beta_1^{t-j} |g_{j,i}| = \sum_{t=1}^{T} \frac{\alpha C}{(1 - \beta_1) \sqrt{\beta_2}} \sum_{i=1}^{d} \sum_{j=1}^{t} \frac{1}{\sqrt{t}} \beta_1^{t-j} |g_{j,i}|$$

$$= \frac{\alpha C}{(1 - \beta_1) \sqrt{\beta_2}} \sum_{i=1}^{d} \sum_{t=1}^{T} |g_{t,i}| \sum_{j=t}^{T} \frac{1}{\sqrt{j}} \beta_1^{j-t} \leq \frac{\alpha C}{(1 - \beta_1) \sqrt{\beta_2}} \sum_{i=1}^{d} \sum_{t=1}^{T} |g_{t,i}| \sum_{j=t}^{T} \frac{1}{\sqrt{t}} \beta_1^{j-t}$$

$$\leq \frac{\alpha C}{(1 - \beta_1)^2 \sqrt{\beta_2}} \sum_{i=1}^{d} \sum_{t=1}^{T} |g_{t,i}| \frac{1}{\sqrt{t}} \leq \frac{\alpha C}{(1 - \beta_1)^2 \sqrt{\beta_2}} \sum_{i=1}^{d} \sqrt{\sum_{t=1}^{T} |g_{t,i}|^2} \sqrt{\sum_{t=1}^{T} \frac{1}{t}}$$

$$\leq \frac{\alpha C}{(1 - \beta_1)^2 \sqrt{\beta_2}} \sum_{i=1}^{d} \sum_{t=1}^{T} |g_{t,i}| \frac{1}{\sqrt{t}} \leq \frac{\alpha C \sqrt{1 + \log T}}{(1 - \beta_1)^2 \sqrt{\beta_2}} \sum_{i=1}^{d} \sqrt{\sum_{t=1}^{T} |g_{t,i}|^2}$$

$$\tag{27}$$

The second equality follows from a re-arrange of sum order. The third inequality comes from the fact that $\sum_{j=t}^{T} \beta_1^{j-t} \leq 1/(1 - \beta_1) \leq 1/(1 - \beta_1)$. The fourth inequality is again from the Cauchy-Schwarz inequality and the final one is from the harmonic sum $\sum_{t=1}^{T} \leq (1 + \log T)$. By denoting $g_{1:t,i}$ to be

the vector of the past graidents from time 1 to $t$ in the $i$-th dimension, i.e. $g_{1:t,i} = [g_{1,i}, g_{2,i}, ..., g_{t,i}]$, we complete the proof of the lemma.

**Lemma A.3** *For the parameter settings and conditions assumed in **Theorem 4.1**, we have*

$$\sum_{t=1}^{T} \frac{1}{\alpha_t} \left[ \|V_t^{1/4}(x_t - x^*)\|^2 - \|V_t^{1/4}(x_{t+1} - x^*)\|^2 \right] \leq \frac{D_\infty^2}{2\alpha_T} \sum_{i=1}^{d} \hat{v}_{T,i}^{1/2} \tag{28}$$

*Proof:* Using the definition of $L2$ norm, by **Lemma A.1**, since $\frac{\hat{v}_{t,i}^{1/2}}{\alpha_t} \geq \frac{\hat{v}_{t-1,i}^{1/2}}{\alpha_{t-1}}$

$$\sum_{t=1}^{T} \frac{1}{\alpha_t} \left[ \|V_t^{1/4}(x_t - x^*)\|^2 - \|V_t^{1/4}(x_{t+1} - x^*)\|^2 \right]$$

$$\leq \frac{1}{\alpha_1} \|V_1^{1/4}(x_1 - x^*)\|^2 + \sum_{t=2}^{T} \left[ \frac{\|V_t^{1/4}(x_t - x^*)\|^2}{\alpha_t} - \frac{\|V_t^{1/4}(x_{t+1} - x^*)\|^2}{\alpha_{t-1}} \right]$$

$$= \frac{1}{\alpha_1} \sum_{i=1}^{d} \hat{v}_{1,i}^{1/2}(x_{1,i} - x_i^*)^2 + \sum_{t=2}^{T} \sum_{i=1}^{d} \left[ \frac{\hat{v}_{t,i}^{1/2}}{\alpha_t}(x_{t,i} - x_i^*)^2 - \frac{\hat{v}_{t-1,i}^{1/2}}{\alpha_{t-1}}(x_{t,i} - x_i^*)^2 \right] \tag{29}$$

$$= \frac{1}{\alpha_1} \sum_{i=1}^{d} \hat{v}_{1,i}^{1/2}(x_{1,i} - x_i^*)^2 + \sum_{t=2}^{T} \sum_{i=1}^{d} \left[ \frac{\hat{v}_{t,i}^{1/2}}{\alpha_t} - \frac{\hat{v}_{t-1,i}^{1/2}}{\alpha_{t-1}} \right](x_{t,i} - x_i^*)^2$$

$$\leq \frac{D_\infty^2}{2\alpha_T} \sum_{i=1}^{d} \hat{v}_{T,i}^{1/2}$$

where the first inequality is from separating the first term and getting rid of the last negative term in the summation. The last inequality is from a telescopic summation and the diameter bound that $\|x - x^*\| \leq D_\infty$

### A.4 Proof of Regret Bound

#### A.4.1 Proof of Theorem 4.1

*Proof.* Following the proof given by Reddi et al. (2018), we provide the proof of regret bound in **Theorem 4.1**. Beginning with the definition of the projection operation $\Pi_{\mathcal{F}, \sqrt{V_t}}$, we have the observation

$$x_{t+1} = \Pi_{\mathcal{F}, \sqrt{V_t}}(x_t - \alpha_t V_t^{-1/2} m_t) = \min_{x \in \mathcal{F}} \|V_t^{1/4}(x - (x_t - \alpha_t V_t^{-1/2} m_t))\| \tag{30}$$

Using Lemma 4 in Reddi et al. (2018) proved by Mcmahan & Streeter (2010) with a direct substitute of $z_1 = (x_t - \alpha_t V_t^{-1/2} m_t), Q = V^{1/2}$ and $z_2 = x^*$ for $x^* \in \mathcal{F}$, the following inequality holds:

$$\|V_t^{1/4}(u_1 - u_2)\|^2 = \|V_t^{1/4}(x_{t+1} - x^*)\|^2 \leq \|V_t^{1/4}(x_t - \alpha_t V_t^{-1/2} m_t - x^*)\|^2$$

$$= \|V_t^{1/4}(x_t - x^*)\|^2 + \alpha_t^2 \|V_t^{-1/4} m_t\|^2 - 2\alpha_t \langle m_t, (x_t - x^*) \rangle$$

$$= \|V_t^{1/4}(x_t - x^*)\|^2 + \alpha_t^2 \|V_t^{-1/4} m_t\|^2 - 2\alpha_t \langle \beta_{1t} m_{t-1} + (1 - \beta_{1t}) g_t, (x_t - x^*) \rangle \tag{31}$$

where the first equality is due to the fact that $\Pi_{\mathcal{F}, \sqrt{V_t}}(x^*) = x^*$. Rearrange the last inequality, we obtain

$$(1 - \beta_{1t})\langle g_t, (x_t - x^*)\rangle \leq \frac{1}{2\alpha_t}\left[\|V_t^{1/4}(x_t - x^*)\|^2 - \|V_t^{1/4}(x_{t+1} - x^*)\|^2\right] + \frac{\alpha_t}{2}\|V_t^{-1/4}m_t\|^2$$
$$- \beta_{1t}\langle m_{t-1}, (x_t - x^*)\rangle$$
$$\leq \frac{1}{2\alpha_t}\left[\|V_t^{1/4}(x_t - x^*)\|^2 - \|V_t^{1/4}(x_{t+1} - x^*)\|^2\right] + \frac{\alpha_t}{2}\|V_t^{-1/4}m_t\|^2$$
$$+ \frac{\beta_{1t}\alpha_t}{2}\|V_t^{-1/4}m_{t-1}\|^2 + \frac{\beta_{1t}}{2\alpha_t}\|V_t^{1/4}(x_t - x^*)\|^2$$

$$(32)$$

The second inequality comes from applications of Cauchy-Schwarz and Young's inequality. We now make use of the approach of bounding the regret using convexify of $f_t$ as in Kingma & Ba (2015). Following **Lemma A.2** and **Lemma A.3**, we have

$$\sum_{t=1}^{T} f_t(x_t) - f_t(x^*) \leq \sum_{t=1}^{T}\langle g_t, (x_t - x^*)\rangle$$
$$\leq \sum_{t=1}^{T} \frac{1}{2\alpha_t(1 - \beta_{1t})}\left[\|V_t^{1/4}(x_t - x^*)\|^2 - \|V_t^{1/4}(x_{t+1} - x^*)\|^2\right] + \frac{\alpha_t}{2(1 - \beta_{1t})}\|V_t^{-1/4}m_t\|^2$$
$$+ \frac{\beta_{1t}\alpha_t}{2(1 - \beta_{1t})}\|V_t^{-1/4}m_{t-1}\|^2 + \frac{\beta_{1t}}{2\alpha_t(1 - \beta_{1t})}\|V_t^{1/4}(x_t - x^*)\|^2$$
$$\leq \frac{D_\infty{}^2}{2\alpha_T(1 - \beta_1)}\sum_{i=1}^{d}\hat{v}_{T,i}^{1/2} + \sum_{t=1}^{T}\frac{\beta_{1t}}{2\alpha_t(1 - \beta_1)}\|V_t^{1/4}(x_t - x^*)\|^2 + \frac{\alpha C\sqrt{1 + \log T}}{(1 - \beta_1)^3\sqrt{\beta_2}}\sum_{i=1}^{d}\|g_{1:T,i}\|_2$$
$$= \frac{D_\infty{}^2}{2\alpha_T(1 - \beta_1)}\sum_{i=1}^{d}\hat{v}_{T,i}^{1/2} + \sum_{t=1}^{T}\frac{1}{2\alpha_t(1 - \beta_1)}\sum_{i=1}^{d}\beta_{1t}(x_{t,i} - x_i^*)^2\hat{v}_{t,i}^{1/2} + \frac{\alpha C\sqrt{1 + \log T}}{(1 - \beta_1)^3\sqrt{\beta_2}}\sum_{i=1}^{d}\|g_{1:T,i}\|_2$$
$$\leq \frac{D_\infty{}^2}{2\alpha_T(1 - \beta_1)}\sum_{i=1}^{d}\hat{v}_{T,i}^{1/2} + \frac{D_\infty^2}{2(1 - \beta_1)}\sum_{t=1}^{T}\sum_{i=1}^{d}\frac{\beta_{1t}\hat{v}_{t,i}^{1/2}}{\alpha_t} + \frac{\alpha C\sqrt{1 + \log T}}{(1 - \beta_1)^3\sqrt{\beta_2}}\sum_{i=1}^{d}\|g_{1:T,i}\|_2$$
$$\leq \frac{D_\infty{}^2}{2\alpha_T(1 - \beta_1)}\sum_{i=1}^{d}\hat{v}_{T,i}^{1/2} + \frac{D_\infty^2}{2(1 - \beta_1)}\sum_{t=1}^{T}\sum_{i=1}^{d}\frac{\beta_{1t}\hat{v}_{t,i}^{1/2}}{\alpha_t} + \frac{\alpha C\sqrt{1 + \log T}}{(1 - \beta_1)^3\sqrt{\beta_2}}\sum_{i=1}^{d}\|g_{1:T,i}\|_2$$

$$(33)$$

### A.4.2 PROOF OF COROLLARY 4.1

*Proof.* We first take a look at the size of $\hat{v}_{t,i}^{1/2}$, note that $\|\nabla f_t(\theta)\|_\infty \leq G_\infty$

$$\hat{v}_{t,i} = \frac{1}{(1 + \beta_{2t})^t - 1}\sum_{j=1}^{t}\beta_{2j}\Pi_{k=1}^{t-j}(1 + \beta_{2(t-k+1)})g_{j,i}^2$$
$$\leq \frac{G_\infty^2}{\beta_2}\sum_{j=1}^{t}\frac{\beta_2}{j}\Pi_{k=1}^{t-j}(1 + \frac{\beta_2}{t - k + 1})$$
$$\leq G_\infty^2\sum_{j=1}^{t}\frac{1}{j}\Pi_{k=1}^{t-j}(1 + \frac{1}{t - k + 1})$$
$$= G_\infty^2\sum_{j=1}^{t}\frac{t + 1}{j(j + 1)} = tG_\infty^2$$

$$(34)$$

The first inequality is due to the fact that $(1 + \beta_{2t})^t \geq (1 + \beta_2)$ and the gradient bound. The second inequality follows from $\beta_2 < 1$. The last inequality is from the telescopic sum. Then we have the following inequality,

$$\sum_{t=1}^{T}\sum_{i=1}^{d}\frac{\beta_{1t}\hat{v}_{t,i}^{1/2}}{\alpha_t} \le dG_\infty \frac{\beta_1}{\alpha}\sum_{t=1}^{T}\lambda^{t-1}t \le \frac{dG_\infty \beta_1}{\alpha(1-\lambda)^2} \tag{35}$$

The second inequality is due to the arithmetic geometric series sum $\sum_{t=1}^{T}\lambda^{t-1}t < \frac{1}{(1-\lambda)^2}$, the reason is as follows

$$\begin{cases} S = \lambda^0 + 2\lambda^1 + \cdots + t\lambda^{t-1} \\ \lambda S = \lambda^1 + 2\lambda^2 + \cdots + t\lambda^t \end{cases} \tag{36}$$

$$(1-\lambda)S = \lambda^0 + \lambda^1 + \cdots + \lambda^{t-1} - t\lambda^t \le \lambda^0 + \lambda^1 + \cdots + \lambda^{t-1} \le \frac{1}{1-\lambda} \tag{37}$$

Therefore we have the following regret bound

$$R_T \le \frac{D_{\infty^2}\sqrt{T}}{2\alpha(1-\beta_1)}\sum_{i=1}^{d}\hat{v}_{T,i}^{1/2} + \frac{d\beta_1 D_\infty^2 G_\infty}{2\alpha(1-\beta_1)(1-\lambda)^2} + \frac{\alpha C\sqrt{1+\log T}}{(1-\beta_1)^3\sqrt{\beta_2}}\sum_{i=1}^{d}\|g_{1:T,i}\|_2 \tag{38}$$

### A.5 PROOF OF NON-CONVEX CONVERGENCE

### A.5.1 PROOF OF THEOREM 4.2

*Proof.* We first directly refer to the original paper and obtain the following bound (Chen et al., 2019).

$$\mathbb{E}\left[\sum_{t=1}^{T}\alpha_t\langle\nabla f(x_t), \nabla f(x_t)/\sqrt{\hat{v}_t}\rangle\right]$$
$$\le \mathbb{E}\left[C_1\sum_{t=1}^{T}\|\alpha_t g_t/\sqrt{v_t}\|^2 + C_2\sum_{t=2}^{T}\|\frac{\alpha_t}{\sqrt{\hat{v}_t}} - \frac{\alpha_{t-1}}{\sqrt{\hat{v}_{t-1}}}\|_1 + C_3\sum_{t=2}^{T-1}\|\frac{\alpha_t}{\sqrt{\hat{v}_t}} - \frac{\alpha_{t-1}}{\sqrt{\hat{v}_{t-1}}}\|^2\right] + C_4 \tag{39}$$

where $C_1, C_2, C_3$ are constants independent of $d$ and $T$, $C_4$ is a constant independent of $T$. For the first term, assume that $\min_{j\in[d]}(\sqrt{\hat{v}_1})_j \ge c > 0$, we have

$$\mathbb{E}\left[C_1\sum_{t=1}^{T}\|\alpha_t g_t/\sqrt{v_t}\|^2\right] \le \mathbb{E}\left[C_1\sum_{t=1}^{T}\|\alpha g_t/c\|^2\right] = \mathbb{E}\left[C_1\sum_{t=1}^{T}\frac{\alpha^2}{c^2}\|g_t\|^2\right] \le \frac{C_1 G_\infty^2 \alpha^2}{c^2} \tag{40}$$

where the first inequality follows from **Lemma A.1** as $\frac{\hat{v}_t}{\alpha_t^2} \ge \frac{\hat{v}_{t-1}}{\alpha_t^2}$. The second inequality is from the gradient bound $\|\nabla f(x_t)\| \le G_\infty$. For the second term with $C_2$, similarly by the positive semi-definiteness in **Lemma A.1**, we have

$$\mathbb{E}\left[C_2\sum_{t=2}^{T}\|\frac{\alpha_t}{\sqrt{\hat{v}_t}} - \frac{\alpha_{t-1}}{\sqrt{\hat{v}_{t-1}}}\|_1\right] = \mathbb{E}\left[C_2\sum_{j=1}^{d}\sum_{t=2}^{T}\left(\frac{\alpha_{t-1}}{(\sqrt{\hat{v}_{t-1}})_j} - \frac{\alpha_t}{(\sqrt{\hat{v}_t})_j}\right)\right]$$
$$= \mathbb{E}\left[C_2\sum_{j=1}^{d}\left(\frac{\alpha_1}{(\sqrt{\hat{v}_1})_j} - \frac{\alpha_T}{(\sqrt{\hat{v}_T})_j}\right)\right] \le \mathbb{E}\left[C_2\sum_{j=1}^{d}\frac{\alpha_1}{(\sqrt{\hat{v}_1})_j}\right] \le \frac{C_2 d\alpha}{c} \tag{41}$$

The second equality is from the telescope sum and for the third term

$$\mathbb{E}\left[C_3 \sum_{t=2}^{T-1} \|\frac{\alpha_t}{\sqrt{\hat{v}_t}} - \frac{\alpha_{t-1}}{\sqrt{\hat{v}_{t-1}}}\|^2\right] \leq \mathbb{E}\left[C_3 \sum_{t=2}^{T-1} \frac{\alpha}{c}\|\frac{\alpha_t}{\sqrt{\hat{v}_t}} - \frac{\alpha_{t-1}}{\sqrt{\hat{v}_{t-1}}}\|_1\right] \leq \frac{C_3 d\alpha^2}{c^2} \tag{42}$$

where the first inequality is because $|\frac{\alpha_t}{\sqrt{v_t}} - \frac{\alpha_{t-1}}{\sqrt{v_{t-1}}}| \leq \frac{\alpha_{t-1}}{\sqrt{v_{t-1}}} \leq \frac{\alpha}{c}$ and the last one is due to the previous inequality with second term. Hence in summary, we have

$$\mathbb{E}\left[\sum_{t=1}^{T} \alpha_t \langle \nabla f(x_t), \nabla f(x_t)/\sqrt{\hat{v}_t} \rangle\right] \leq \frac{C_1 G_\infty^2 \alpha^2}{c^2} + \frac{C_2 d\alpha}{c} + \frac{C_3 d\alpha^2}{c^2} + C_4 \tag{43}$$

Now by the proof of **Corollary 4.1**, we have an upper bound $(\hat{v}_t)_j \leq tG_\infty^2$, therefore we can bound the effective step sizes,

$$\frac{\alpha_t}{(\sqrt{v_t})_j} \geq \frac{\alpha}{tG_\infty} \tag{44}$$

And thus we have

$$\mathbb{E}\left[\sum_{t=1}^{T} \alpha_t \langle \nabla f(x_t), \nabla f(x_t)/\sqrt{\hat{v}_t} \rangle\right] \geq \mathbb{E}\left[\sum_{t=1}^{T} \frac{\alpha}{tG_\infty} \|\nabla f(x_t)\|^2\right] \geq \frac{\alpha}{G_\infty} \min_{t\in[T]} \mathbb{E}\left[\|\nabla f(x_t)\|^2\right] \sum_{t=1}^{T} \frac{1}{t}$$
$$\geq \frac{\alpha}{G_\infty} \min_{t\in[T]} \mathbb{E}\left[\|\nabla f(x_t)\|^2\right] \log(1+T) \tag{45}$$

The last inequality is due to the fact that $\sum_{t=1}^{T} 1/t \geq \int_{t=1}^{T+1} 1/t \, dt = \log(1+T)$, therefore we have

$$\min_{t\in[T]} \mathbb{E}\left[\|\nabla f(x_t)\|^2\right] \leq \frac{G_\infty}{\alpha \log(1+T)} \left(\frac{C_1 G_\infty^2 \alpha^2}{c^2} + \frac{C_2 d\alpha}{c} + \frac{C_3 d\alpha^2}{c^2} + C_4\right) \tag{46}$$

We would emphasize that the assumption $\|\alpha_t m_t/\sqrt{\hat{v}_t}\| \leq G$ in the theorem is automatically satisfied as $\frac{\alpha_t}{\sqrt{\hat{v}_t}} \leq \frac{\alpha_1}{\sqrt{\hat{v}_1}} = \frac{\alpha}{c}$. Hence $\|\alpha_t m_t/\sqrt{\hat{v}_t}\| \leq \frac{\alpha G_\infty}{c}$.

### A.5.2 PROOF OF COROLLARY 4.2

Similar to the proof of **Theorem 4.2**, we still have

$$\mathbb{E}\left[\sum_{t=1}^{T} \alpha_t \langle \nabla f(x_t), \nabla f(x_t)/\sqrt{\hat{v}_t} \rangle\right] \leq \frac{C_1 G_\infty^2 \alpha^2}{c^2} + \frac{C_2 d\alpha}{c} + \frac{C_3 d\alpha^2}{c^2} + C_4 \tag{47}$$

Note that $(\hat{v}_t)_j$ has the following upper bound as $\|\nabla f_t(\theta)\|_\infty \leq G_\infty$,

$$\hat{v}_{t,i} = \frac{1}{(1+\beta_2)^t - 1} \sum_{j=1}^{t} \beta_2 (1+\beta_2)^{t-j} g_{j,i}^2$$
$$\leq \frac{G_\infty^2}{(1+\beta_2)^t - 1} \sum_{j=1}^{t} \beta_2 (1+\beta_2)^{t-j} = G_\infty^2 \tag{48}$$

And thus we have

$$\mathbb{E}\left[\sum_{t=1}^{T}\alpha_t\langle\nabla f(x_t), \nabla f(x_t)/\sqrt{\hat{v}_t}\rangle\right] \geq \mathbb{E}\left[\sum_{t=1}^{T}\frac{\alpha}{\sqrt{t}G_\infty}\|\nabla f(x_t)\|^2\right] \geq \frac{\alpha}{G_\infty}\min_{t\in[T]}\mathbb{E}\left[\|\nabla f(x_t)\|^2\right]\sum_{t=1}^{T}\frac{1}{\sqrt{t}}$$

$$\geq \frac{\alpha}{G_\infty}\min_{t\in[T]}\mathbb{E}\left[\|\nabla f(x_t)\|^2\right]\sqrt{T}$$

$$(49)$$

where the last inequality is by the fact that $\sum_{t=1}^{T}\frac{1}{\sqrt{t}} \geq \sqrt{T}$, therefore we have

$$\min_{t\in[T]}\mathbb{E}\left[\|\nabla f(x_t)\|^2\right] \leq \frac{G_\infty}{\alpha\sqrt{T}}(\frac{C_1 G_\infty^2 \alpha^2}{c^2} + \frac{C_2 d\alpha}{c} + \frac{C_3 d\alpha^2}{c^2} + C_4) \qquad (50)$$

## A.6 IMPLEMENTATION OF ADAX AND ADAX-W

The detailed implementations of AdaX and AdaX-W are as follows. The performance of AdaX is robust with respect to the value of $\beta_2$, but we recommend smaller values such as $1e-4, 1e-5$ to reduce computational cost. The value of $\beta_3$ can be set as $0.9$ to match the first moment design of Adam, consequently we need $\frac{1}{100}$ step sizes. The size of weight decay should also be adjusted respectively.

---

**Algorithm 3** AdaX Algorithm with $L_2$ Regularization and Weight Decay

1: **Input:** $x \in \mathcal{F}$, step size $\{\alpha_t\}_{t=1}^{T}$, $(\beta_1, \beta_2, \beta_3) = (0.9, 1e-4, 0.999)$, weight decay $\lambda = 5e-4$, $\epsilon = 1e-12$
2: **Initialize** $m_0 = 0, v_0 = 0$
3: **for** $t = 1$ **to** $T$ **do**
4: $\quad g_t = \nabla f_t(x_t) + \lambda x_t$
5: $\quad m_t = \beta_1 m_{t-1} + (1-\beta_3)g_t$
6: $\quad v_t = (1+\beta_2)v_{t-1} + \beta_2 g_t^2$
7: $\quad \hat{v}_t = v_t/[(1+\beta_2)^t - 1]$ and $V_t = \text{diag}(\hat{v}_t)$
8: $\quad x_{t+1} = \Pi_{\mathcal{F},\sqrt{V_t}}(x_t - \alpha_t(m_t/(\sqrt{\hat{v}_t} + \epsilon) + \lambda x_t))$
9: **end for**

---

## A.7 HYPER-PARAMETER TUNING IN THE EXPERIMENTS

The hyperparameters in different algorithms have a huge impact on their performances in the experiments. To efficiently find the best step sizes, we follow Wilson et al. (2017) to perform a logarithmically-spaced grid search of the optimal step sizes and we list the step sizes we tried in the following tables, where the step sizes in bold are the ones with best performances and used in the experiments section.

**Step size: Image Classification**

- **SGD(M)** $\{10, 1, \mathbf{1e\text{-}1}, 1e\text{-}2, 1e\text{-}3\}$

- **Adam/AdamW** $\{1e\text{-}2, 3e\text{-}3, \mathbf{1e\text{-}3}, 3e\text{-}4, 1e\text{-}4\}$

- **AdaGrad** $\{5e\text{-}2, \mathbf{1e\text{-}2}, 5e\text{-}3, 1e\text{-}3, 5e\text{-}4\}$

- **AMSGrad** $\{1e\text{-}2, 3e\text{-}3, \mathbf{1e\text{-}3}, 3e\text{-}4, 1e\text{-}4\}$

- **RMSProp** $\{1e\text{-}2, 5e\text{-}3, \mathbf{1e\text{-}3}, 5e\text{-}4, 1e\text{-}4\}$

- **AdaX(ours)** $\{5e\text{-}1, 2.5e\text{-}1, 2e\text{-}1, \mathbf{1.5e\text{-}1}, 1e\text{-}1, 5e\text{-}2\ 1e\text{-}2, 5e\text{-}3, 1e\text{-}3\}$

- **AdaX-W(ours)** $\{1, \mathbf{5e\text{-}1}, 4e\text{-}1, 3e\text{-}1, \mathbf{2.5e\text{-}1(CIFAR)}, 1e\text{-}1, 1e\text{-}2, 5e\text{-}3, 1e\text{-}3\}$

**Step size: VOC2012 Segmentation**

- **SGD(M)** {1e-3, 5e-4, **2.5e-4**, 1e-4, 5e-5}

- **AdamW** {5e-4, 1e-5, 5e-5, **1e-6**, 5e-7}

- **AMSGrad** {5e-4, 1e-5, 5e-5, **1e-6**, 5e-7}

- **RMSProp** {1e-4, 5e-5, **1e-5**, 5e-6, 1e-6}

- **AdaX-W(ours)** {1e-2, 5e-3, **1e-3**, 5e-4, 1e-4}

**Step size: Billionwords**

- **SGD** {30, **20**, 10, 5, 1, 1e-1, 1e-2 }

- **Adam** {5e-3, 2e-3, 1e-3, **5e-4**, 1e-4 }

- **AdaX(ours)** {5e-3, 2e-3, 1e-3, **5e-4**, 1e-4} × {0.5, 1, 5, **15**, 25, 50, 100}

**Momentum parameters.** For the momentum parameters of Adam, RMSProp and AMSGrad, we tuned over $(\beta_1, \beta_2) = \{(0.9, 0.999), (0.99, 0.999), (0.99, 0.9999)\}$ and found that the default values $(0.9, 0.999)$ as in Kingma & Ba (2015) yield the best result. For the momentum parameters $(\beta_1, \beta_2, \beta_3)$ in AdaX, we directly applied $\beta_1 = 0.9$ as in Adam and we tuned $\beta_2$ over $\{1e - 3, 1e - 4, 1e - 5\}$. We found that the value of $\beta_2$ didn't affect the general performance of AdaX. Tuning $\beta_3$ is the same as tunning the step sizes, as we will show in the next subsection.

For the other hyperparameters such as weight decay and batch size, we directly apply the same settings as in the baselines (He et al., 2016)(Chen et al., 2016)(Rdspring1) (Loshchilov & Hutter, 2019).

## A.8   THE EFFECT OF $\beta_3$

As we mentioned in section 4 and 5, adding a new hyper-parameter $\beta_3$ does not influence the first momentum sum of AdaX as it is just a constant common factor that can be extracted out as follows.

$$m_t = \beta_1 m_{t-1} + (1 - \beta_3)g_t = (1 - \beta_3)\sum_{i=1}^{t} \beta_1^{t-i} g_i \tag{51}$$

Therefore, tuning the hyperparameter $\beta_3$ will have exactly the same effect as tuning the step size $\alpha$. As long as $(1 - \beta_3)\alpha$ remains the same, the performance of AdaX doesn't change. We also show this fact empirically using different combinations of $(\alpha, \beta_3)$ on the CIFAR-10 dataset.

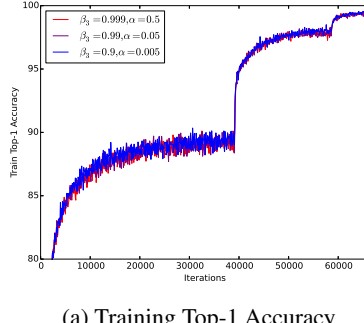
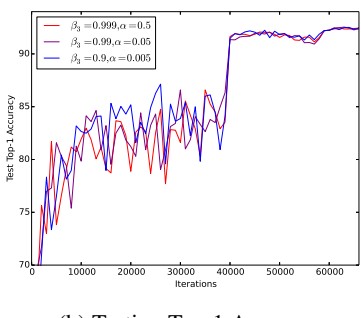

(a) Training Top-1 Accuracy                    (b) Testing Top-1 Accuracy

Figure 5: Training and Testing Results on CIFAR-10 with Different combinations of $(\alpha, \beta_3)$.

As shown above, we trained the ResNet-20 model (He et al., 2016) on the CIFAR-10 dataset with different combinations of $\alpha$ and $\beta_3$ from $\{(0.5, 0.999), (0.05, 0.99), (0.005, 0.9)\}$. However, the

value of $(1 - \beta_3)\alpha$ is kept to be 5e-4. It can be easily observed that they have approximately the same training and testing curves, as well as similar final performances, which proves our claim.

