# OpenReview forum: "AdaX: Adaptive Gradient Descent with Exponential Long Term Memory"
_ICLR.cc/2020/Conference — Reject_

### Official Review · AnonReviewer3 · 2019-10-11
**Official Blind Review #3**

**Rating:** 1

**Review:**

This paper introduces a new step-size adaptation algorithm called AdaX. AdaX
builds on the ideas of the Adam algorithm to address instability and non-convergence issues.
Convergence of AdaX is proven in both convex and non-convex settings. The paper
also provides an empirical comparison of AdaX against its predecessors
(SGD, RMSProp, Adam, AMSGrad) on a variety of tasks.

I recommend the paper be rejected. I believe the convergence results could be a significant contribution, but the
quality of the paper is hampered by its experimental design. The paper felt generally unpolished, containing
frequent grammatical errors, imprecise language, and uncited statements.

My main issue with the paper is the experimental design. I am not convinced that we can
draw valid conclusions from the experimental results for the following reasons:
  - The experiments are lacking important details. How many independent runs of the experiment
    were the experimental results averaged over? All of the experiments have random initial conditions
    (e.g. initialization of the network), and should be ran multiple times, not just once.
    There's no error bars in any of the plots, so it's unclear whether AdaX really
    does provide a statistically significant improvement over the baselines.
    Similarly, the data in all the tables is quite similar, so without indicating the
    spread of these estimates its impossible to tell whether these results are significant or not.

  - How were the hyperparameters and step-size schedules chosen? The performance of Adam, AMSGrad, and
    RMSProp are quite sensitive to their hyperparameters, and the optimal hyperparameters are problem-dependent.
    Some of the experiments just use the default hyperparameters; this is insufficient when trying to directly
    compare the performance of these methods, as their performance can vary greatly with different values of
    these parameters. I'm not convinced that we should be drawing conclusions about the relative
    performance of these algorithms from any of the experiments for this reason.

Of course, meaningful empirical results are not necessarily characterized by statistically outperforming the baselines.
Well designed experiments can highlight important ways in which the performances differ, providing the community
with a deeper understanding of the methods investigated. I would argue that the experiments in the paper do not
achieve this either; the experiments do not provide any new intuition or understanding of the methods, showing
only the relative performances in terms of learning curves on a somewhat random collection of supervised learning problems. Why were these specific problems chosen? What makes these problems ideal for showcasing the performance of AdaX? If AdaX is an improvement over Adam, why? What exactly is happening with it's effective step-sizes that leads
to the better performance? Can you show how their step-sizes differ over time?

Statements that need citation or revision:
  - "Adaptive optimization algorithms such as RMSProp and Adam... as well as weak performance
     compared to the first order gradient methods such as SGD" (Abstract). This needs a citation.
     Similarly, "AdaX outperforms various tasks of computer vision and natural language processing and can catch
     up with SGD"; as above, I'm unaware of work (other than theoretical) that shows that SGD significantly
     outperforms Adam in deep neural networks.
  -  "In the era of deep learning, SGD ... remains the most effective algorithm in training deep neural
      networks" (Introduction). What are you referring to here? Vanilla SGD? Or are you including Adam etc here?
      As above, this should have a citation. Adam's popularity is largely due to its effectiveness in training
      deep neural networks.
  - "However, Adam has worse performance (i.e. generalization ability in testing stage) compared with SGD"
     (Introduction). Citation needed.
  - In the last paragraph of the Introduction, you introduced AdaX twice: "To address the above issues, we propose a
    new adaptive optimization method, termed AdaX, which guarantees convergence...", and, "To address
    the above problems, we introduce a novel AdaX algorithm and theoreetically prove that it converges..."


**Experience Assessment:**

I have published one or two papers in this area.

**Review Assessment: Checking Correctness Of Derivations And Theory:**

I assessed the sensibility of the derivations and theory.

**Review Assessment: Checking Correctness Of Experiments:**

I carefully checked the experiments.

**Review Assessment: Thoroughness In Paper Reading:**

I read the paper thoroughly.

---

> ### Author Response · Authors · 2019-11-09
> **Rebuttal: about the "problems"**
>
> $>>>$ The experiments are lacking important details. How many independent runs of the experiment
> were the experimental results averaged over? All of the experiments have random initial conditions
> (e.g. initialization of the network), and should be ran multiple times, not just once.
>
> Our results are averaged over multiple independent runs (e.g. more than 5 runs for CIFAR and 3 for ImageNet). However, thank you for pointing it out. We will consider adding error bars to our experiments if time allows.
>
> $>>>$ How were the hyperparameters and step-size schedules chosen? The performance of Adam, AMSGrad, and RMSProp are quite sensitive to their hyperparameters, and the optimal hyperparameters are problem-dependent.
>
> We have carefully and thoroughly tuned the parameters and we will add our tuning strategy to our paper as soon as possible. The results in the paper are all the best results we are able to obtain.
>
> $>>>$ The experiments do not provide any new intuition or understanding of the methods, showing
> only the relative performances in terms of learning curves on a somewhat random collection of supervised learning problems. Why were these specific problems chosen? What makes these problems ideal for showcasing the performance of AdaX? If AdaX is an improvement over Adam, why? What exactly is happening with its effective step-sizes that leads
> to the better performance? Can you show how their step-sizes differ over time?
>
> These experiments are definitely not a random collection of supervised learning problems. They cover the most heated research topics in deep learning, i.e. computer vision, natural language processing, and transfer learning. Many optimization papers have chosen these tasks to show the superiority of their algorithms, such as [1]-[5]. We will show the effective step sizes of different algorithms in our toy example (Problem (3)) in the future versions of our paper, but the difference of their step sizes can be easily observed from the training curves in the experiments. AdaX outperforms Adam because Adam converges too fast and may end up in local minimums and we mentioned this reason in our section 3 and 4. The experiments verify our claim by showing the slightly slower convergence speed but far better performances. We are strongly against your statement that the experiments do not provide any new intuition or understanding of the methods and are just a random collection.
>
> $>>>$ Adaptive optimization algorithms such as RMSProp and Adam... as well as weak performance
>      compared to the first order gradient methods such as SGD" (Abstract). This needs a citation
>
> We believe that it is recommended not to cite other people's work in the abstract, and such a convention can be observed from this year's papers as well, for example (https://openreview.net/pdf?id=S1xJ4JHFvS), (https://openreview.net/pdf?id=rye5YaEtPr) and (https://openreview.net/pdf?id=rkgz2aEKDr).
>
> $>>>$I'm unaware of work (other than theoretical) that shows that SGD significantly outperforms Adam in deep neural networks.
>
> There are many papers (both theoretical and experimental) that show SGD(M) significantly outperforms Adam in deep neural networks, for example [1], [3], and [4]. As mentioned by reviewer 2 and reviewer 4, the poor generalization of adaptive methods is one of the main reasons why people develop so many different algorithms.
>
> We apologize for the other missing citations and will fix those as soon as possible.
>
> In conclusion, we thank the reviewer for suggesting error bars, parameter tuning, and some missing citations, but we do not agree with the other "problems" in our paper mentioned in the review. We hope the AC and reviewer can reconsider this evaluation.
>
>
> [1]Haiwen Huang, Chang Wang, and Bin Dong. Nostalgic adam: Weighting more of the past gradients
> when designing the adaptive learning rate. arXiv preprint arXiv: 1805.07557, 2019.
>
> [2]Diederik P Kingma and Jimmy Lei Ba. Adam: A method for stochastic optimization. Proceedings
> of the 3rd International Conference on Learning Representations (ICLR), 2015.
>
> [3]Ilya Loshchilov and Frank Hutter. Decoupled weight decay regularization. Proceedings of 7th
> International Conference on Learning Representations (ICLR), 2019.
>
> [4]Liangchen Luo, Yuanhao Xiong, Yan Liu, and Xu Sun. Adaptive gradient methods with dynamic
> bound of learning rate. Proceedings of 7th International Conference on Learning Representations,
> 2019.
>
> [5]Sashank J. Reddi, Stayen Kale, and Sanjiv Kumar. On the convergence of adam and beyond.

---

### Official Review · AnonReviewer1 · 2019-10-24
**Official Blind Review #1**

**Rating:** 3

**Review:**

In this paper, the authors propose a new adaptive gradient algorithm AdaX, which the authors claim results in better convergence and generalization properties compared to previous adaptive gradient methods. The paper overall is fairly clear, although the writing can be improved in places.

Section 3 is interesting, although calling the section the "nonconvergence of Adam" is a bit misleading, since the algorithm does converge to a local minimum.

I have some concerns about the rest of the paper however. I am a bit confused about the changes proposed to Adam that gives rise to the AdaX algorithm.

1. Won't replacing beta2 with 1+beta2 keep increasing the magnitude of the denominator of the algorithm just like AdaGrad does? In that case, is the algorithm expected to work better when having sparse gradients?

2. I also do not quite understand how to interpret using a separate hyperparameter for the momentum term (ie the beta3 hyperparameter that is introduced). How is beta1 and beta3 related? The numerator loses the interpretation of momentum, i.e., averaged past gradients, when using a separate beta3 parameter, and this does not feel like a principled change.

I have a number of questions about the experiments as well, which makes it hard for me to interpret the significance of the empirical results presented:

1. What is the minibatch size used? Do any of the conclusions presented change if the minibatch size is changed?

2. Is the learning rate tuned? The authors mention the initial learning rate used for the experiments, but it is not clear why those values are used? Was the same initial learning rate used for all algorithms?

3. Were the beta1, beta2 and beta3 values tuned for AdaX? What about beta1 and beta2 for Adam? What are the optimal values of these parameters that were observed?

4. How sensitive is performance to the values of these hyperparameters?

Overall I think this work requires quite a bit of work before it is ready for publication, and would benefit from a much more thorough empirical evaluation of the algorithm.

==================================

Edit after rebutall:
I thank the authors for their response. While the paper has definitely improved in the newer draft, after having read the other reviews and the updated draft, I believe the paper still requires a bit of work before being ready for publication. I am sticking to my score.

**Experience Assessment:**

I have published in this field for several years.

**Review Assessment: Checking Correctness Of Derivations And Theory:**

I assessed the sensibility of the derivations and theory.

**Review Assessment: Checking Correctness Of Experiments:**

I carefully checked the experiments.

**Review Assessment: Thoroughness In Paper Reading:**

I read the paper at least twice and used my best judgement in assessing the paper.

---

> ### Author Response · Authors · 2019-11-09
> **Rebuttal: about your questions and concerns**
>
> Thank you for your interest in our paper. We hope that we can resolve your concerns in this rebuttal.
>
> $>>>$ Won't replacing beta2 with 1+beta2 keep increasing the magnitude of the denominator of the algorithm just like AdaGrad does? In that case, is the algorithm expected to work better when having sparse gradients.
>
> The magnitude of $v_t$ does not necessarily keep increasing because of the existence of the bias correction term$(1+\beta_2)^t-1$. Therefore our algorithm is totally different from the AdaGrad algorithm. However with a strongly decreasing step size $\alpha_t = \alpha/\sqrt{t}$ as in the theory proofs, $v_t/\alpha_t$ does keep increasing and this fact corresponds to our claim that the matrix $\Gamma_t = \frac{\sqrt{V_{t+1}}}{\alpha_{t+1}}-\frac{\sqrt{V_{t}}}{\alpha_t}$ is positive definite in our algorithm. We tested the performance of AdaX on the one-billion-word dataset where sparse gradients existed and our algorithm did perform even better than Adam.
>
> $>>>$I also do not quite understand how to interpret using a separate hyperparameter for the momentum term (ie the beta3 hyperparameter that is introduced). How is beta1 and beta3 related?
>
> We have found that this new parameter $\beta_3$ makes our reviewers confused and we are genuinely sorry for the confusion. The $\beta_3$ parameter is introduced mainly to help proving the theoretical convergence results as in Theorem 4.1. However, it does not change the interpretation of momentum at all because the constant $(1-\beta_3)$ can be factored out of each term of the first-order momentum sum. As long as $\alpha (1-\beta_3)$ does not change, the effective update steps of AdaX remain exactly the same. We will include more experiments to show this fact in future versions of our paper. Thank you for pointing it out.
>
> $>>>$ What is the minibatch size used? Do any of the conclusions presented change if the minibatch size is changed?
>
> As mentioned in our paper, the mini-batch sizes used in our experiments are the same as those used in the original papers and studies, i.e. 128 for CIFAR-10 and 256(32*8) for ImageNet as in [1], 10 for VOC2012 as in [2], and 128 for One-Billion word as in [3]. We tried changing the batch sizes in our experiments, but the conclusions don't change.
>
> $>>>$  Is the learning rate tuned? The authors mention the initial learning rate used for the experiments, but it is not clear why those values are used? Was the same initial learning rate used for all algorithms?
>
> Yes, we have tuned the learning rates for different optimizers. The values we choose are the parameters that yield the best results in our experiments, and they are the same as those in the original papers [1][2][3]. As we mentioned in the experiments section, different initial step sizes should be used for different optimizers due to their different designs. For example in image classification tasks, Adam is known to work well with initial step size from 1e-4 to 1e-3 while SGD needs much larger initial step sizes, such as 0.1.
>
> $>>>$ Were the beta1, beta2 and beta3 values tuned for AdaX? What about beta1 and beta2 for Adam? What are the optimal values of these parameters that were observed?
>
> Yes, we have tuned the $\beta_1$ and $\beta_2$ values in our experiments. $\beta_1 = 0.9$ generates the best results when the other parameters are held constant. We found that the value of $\beta_2$ can be chosen from $1e-3$ to $1e-5$ and it does not affect the experiment results. We choose $1e-4$ only to avoid the large computation cost of large numbers. Tuning $\beta_3$ is the same as tuning the step sizes $\alpha_t$ as mentioned earlier, and we will show this fact by conducting more experiments. We have also tuned the $\beta_1$ and $\beta_2$ parameters for Adam, and we found that the default parameters recommended by [4] (i.e. $\beta_1=0.9, \beta_2=0.999$) performed the best.
>
> $>>>$ How sensitive is performance to the values of these hyperparameter?
>
> In AdaX, the overall performance is not sensitive to $\beta_2$ as mentioned earlier, but it is sensitive to $\beta_1$ and we found that $\beta_1 =0.9$ yields the best performances. The parameter $\beta_3$ can be changed and the performance is not affected as long as $(1-\beta_3)\alpha$ remains constant.
>
> Again, thank you for your interesting questions and suggestions about our paper. We will add more experiments to our paper very soon.
>
> [1] He et al. Deep residual learning for image recognition. Proceedings of the IEEE Conference on Computer Vision and Pattern Recognition (CVPR),2016
> [2] Chen et al.Deeplab: Semantic image segmentation with deep convolutional nets, atrous convolution, and
> fully connected crfs. IEEE Transactions on Pattern Analysis and Machine Intelligence, 40:834–848, 2016.
> [3] Rdspring1. Pytorch gbw lm. https://github.com/rdspring1/PyTorch_GBW_LM.
> [4] Kingma and Ba. Adam: A method for stochastic optimization. Proceedings of the 3rd International Conference on Learning Representations (ICLR), 2015.

---

### Official Review · AnonReviewer4 · 2019-10-24
**Official Blind Review #4**

**Rating:** 3

**Review:**

This paper proposed a new adaptive gradient descent algorithm with exponential long term memory. The authors analyzed the non-convergence issue in Adam into a simple non-convex case. The authors also presented the convergence of the proposed AdaX in both convex and non-convex settings.

- The proposed algorithm revisited the non-convergence issue in Adam and proposed a new algorithm design to try to address this issue. However, the new algorithm design is a bit strange to me, especially in Line 6 of Algorithm 2, the authors proposed to update v_t by (1+ \beta_2) v_{t-1} + \beta_2 g_t^2, where normally people would use (1-\beta2) and \beta2 as the coefficients. I am not quite get the intuition of using such as strange design. I wonder if the authors could further explain that.

- The authors also add change \beta_1 in Line 5 of Algorithm 2 into \beta_3. And then in theory, the authors again choose \beta_3 as \beta_1. It seems that \beta_3 is not contributing to any theoretical result. It seems to me to just have another parameter to tune in order to get better performances. Can the authors gives more justification on why introducing such a term here? And also show how the different choice of \beta_3 affects the final result?

- Missing some important references closely related to this paper:

Chen, Jinghui, and Quanquan Gu. "Closing the generalization gap of adaptive gradient methods in training deep neural networks." arXiv preprint arXiv:1806.06763 (2018).
Zaheer, Manzil, et al. "Adaptive methods for nonconvex optimization." Advances in Neural Information Processing Systems. 2018.
Zhou, Dongruo, et al. "On the convergence of adaptive gradient methods for nonconvex optimization." arXiv preprint arXiv:1808.05671 (2018).

I would suggest the authors to also compare with the above mentioned baselines to better demonstrate its performances and theoretical results.

- The authors include theoretical analysis in both convex and non-convex settings, which is appreciated, however, the theoretical result seems to show similar convergence guarantees with AMSGrad. I wonder if the authors could provide theoretical justifications on why the proposed method is better than prior arts, probably some sharper convergences or some generalization guarantees?

- In the experiments part, I wonder why the authors did not compare with AMSGrad, RMSProp in later parts such as ImageNet, IoU and RNN parts? I makes no sense to drop them for those experiments. Also, are the authors fully tuned the hyper-parameters for other baselines such as step size and weight decay on SGDM?

================
after the rebuttal

I thank the authors for their response but I still feel that the intuition of this paper is not clear enough and comparison with more baselines is needed. Therefore I decided to keep my score unchanged.

**Experience Assessment:**

I have published in this field for several years.

**Review Assessment: Checking Correctness Of Derivations And Theory:**

I carefully checked the derivations and theory.

**Review Assessment: Checking Correctness Of Experiments:**

I carefully checked the experiments.

**Review Assessment: Thoroughness In Paper Reading:**

I read the paper thoroughly.

---

> ### Author Response · Authors · 2019-11-07
> **Rebuttal: about the questions and the suggestions**
>
> Thank you for your interest in our paper and your constructive feedback! We want to clarify the following things about our paper.
>
> $>>>$ However, the new algorithm design is a bit strange to me, especially in Line 6 of Algorithm 2, the authors proposed to update $v_t$ by $(1+ \beta_2) v_{t-1} + \beta_2 g_t^2$, where normally people would use $(1-\beta_2)$ and $\beta_2$ as the coefficients. I am not quite get the intuition of using such as strange design. I wonder if the authors could further explain that.
>
> The change from exponential moving average($\beta_2 v_{t-1} + (1-\beta_2) g_t^2$) to exponential long term memory($(1+ \beta_2) v_{t-1} + \beta_2 g_t^2$) is the most important reason why our algorithm outperforms Adam. As shown in our toy example (3), because Adam assigns high weights on recent gradients and low weights on past gradients, its second momentum decreases with approximately the same rate as the first momentum. As a result, the effective update steps of Adam ignore the gradient decrease information and will be larger than a constant. Adam converges very fast but will ultimately be trapped in the local minimum. Therefore, we conclude that current gradients are not trustworthy and past information should not be forgotten as in Adam. We change the moving average design to exponential long term memory in order to emphasize the importance of past memory (high weight $(1+\beta_2)$) and assign less weight ($\beta_2$) on current gradients. Such a design slows down the convergence speed of Adam a little, but results in better performances in our toy example as the gradient decrease information is not lost. In our experiment section, we also observe that AdaX converges slightly slower than Adam at the beginning of different tasks, but its final performance can catch up with SGDM.
>
> $>>>$The authors also add change $\beta_1$ in Line 5 of Algorithm 2 into $\beta_3$. And then in theory, the authors again choose $\beta_3$ as $\beta_1$. It seems that $\beta_3$ is not contributing to any theoretical result. It seems to me to just have another parameter to tune in order to get better performances. Can the authors gives more justification on why introducing such a term here? And also show how the different choice of $\beta_3$ affects the final result?
>
> We have found that the new hyperparameter $\beta_3$ has caused some confusion and we want to apologize for that. However, we want to clarify that $\beta_3$ does help to prove the theoretical conclusion in Theorem 4.1 as we state $\beta_{3t} = 1-1/\sqrt{t}$ in the assumptions. The reason that $\beta_3$ is changed to $\beta_1$ in Theorem 4.2 is to match up with the settings of the original paper(Chen et. al.  On the convergence of a class of adam-type algorithm for non-convex optimization, 2019). Besides, $(1-\beta_3)$ only scales the effective update steps by a constant and tuning this parameter is the same as tuning the step size $\alpha$. Therefore the better performance of AdaX does not originate from this new parameter. We will add more experiments in future versions of our paper to show that as long as $\alpha(1-\beta_3)$ remains the same, the performance of AdaX does not change.
>
> $>>>$I would suggest the authors to also compare with the above mentioned baselines to better demonstrate its performances and theoretical results.
>
> We will consider adding these baselines in theory and experiments to show our superiority if time allows. Thank you for bringing them up.
>
> $>>>$In the experiments part, I wonder why the authors did not compare with AMSGrad, RMSProp in later parts such as ImageNet, IoU and RNN parts? I makes no sense to drop them for those experiments. Also, are the authors fully tuned the hyper-parameters for other baselines such as step size and weight decay on SGDM
>
> We didn't compare with AMSGrad, RMSProp in the later experiments because their performances are generally close to or even worse than Adam (as shown in the CIFAR experiments). Because our method can catch up with SGDM in so many different tasks, we are confident that our method will outperform these methods. However, we will consider adding their performances in later versions of our paper. Thank you for your suggestion. We have definitely fully tuned the hyper-parameters of other baselines, especially for Adam and SGD. All the shown experiments are the best results we are able to obtain and we will add our tuning strategies to the Appendix.
>
> Again, thank you for your feedback and we hope our rebuttal can address your concerns.

---

### Official Review · AnonReviewer2 · 2019-10-26
**Official Blind Review #2**

**Rating:** 3

**Review:**

This paper points out that existing adaptive methods (especially for the methods designing second-order momentum estimates in a exponentially moving average fashion) do not consider gradient decrease information and this might lead to suboptimal convergences via simple non-cvx toy example.
Based on this observation, the authors provide a novel optimization algorithm for long-term memory of past gradients by modifying second-order momentum design. Also, they provide aconvex regret analysis and convergence analysis for non-convex optimization. Finally, the authors evaluate their methods on various deep learning problems.

Significance/Novelty: While there have been many studies on non-convergence of Adam, raising an issue on ignoring the gradient decrease information seems novel.

Pros:
1. The motivating toy example in Section 3 is useful for readers to get intuitions.

2. By introducing long-term memory on past-gradients, the authors fix the Adam's issues and they can also improve the convergence rate in a non-convex optimization (Corollary 4.2).

3. Empirical studies show superiority to original Adam (Section 5).

Cons:
While they provide a significant study on Adam's failure and a novel optimization algorithm, I have several concerns:
1. What is default hyperparameters for AdaX in Algorithm 1? Is it the same as AdaX-W (Algorithm 3) in Appendix? The bias correction term in the line 7 of Algorithm 1 will be very large even with small $\beta_2$ since it is expoential (For example, (1 + 0.0001)^(100000) ~ 20000 for $\beta_2$ = 10^(-4)). So, it is not clear that the second momentum estimate of Ada-X is really stable. For this, it would be interesting to see how the trajectories of second-order momentum estimates of Adam, AMSGrad, Ada-X are different. I think this will help to understand Ada-X better.

2. In terms of theory, I think the Lemma 4.1 is inevitable for convergence guarantees in Theorem 4.1 and Theorem 4.2. Although the authors effectively remove log T in the numerator in Corollary 3.2 of Chen et al. (2019) using their lemma 4.1 (I think this is the key point), the assumption that $\beta_{2t} = \beta_2 / t$ seems quite strong, and original Adam paper has no such assumptions. For a real deep learning problems such as training ResNet on CIFAR-10, the $\beta_{2t}$ is almost zero after even one or two epochs where Ada-X behaves like vanilla SGD. Is there no room for relaxing this assumption such as $\beta_{2t} = \beta_2 / \sqrt{t}$? Also, it is not clear how the authors derive Corollary 4.2 from Theorem 4.2 since Theorem 4.2 assumes $\beta_{2t} = \beta_2 / t$ while Corollary 4.2 does not.

3. In the experiment, it is not clear that the authors use the same strategy for constructing first-order momentum for Adam with a newly introduced parameter $\beta_3$. In other words, the authors should use the same policy on constructing the first-order momentum estimate for both Adam and Ada-X. Also, as the authors add an additional hyperparameter $\beta_3$, the effect of $\beta_3$ on performance should be discussed at least empirically.

4. There are many studies on fixing poor generalization of adaptive methods (such as AdaBound which the authors cited). In this context, Zaheer et al. (2018, Adaptive methods for non-convex optimization) propose a large epsilon value (numerical stability parameter) such as $\epsilon = 10^{-3}$ for better generalization. It will be more interesting to see the comparisons in this regime.

5. In my experience with Adam-W (Decoupled weight decay regularization), Adam-W requires a relatively large weight decay parameter $\lambda$.
As an example, DenseNet-BC-100-12 shows a similar validation accuracy with Adam-W $\lambda = 0.05$ under the learning rate scheduling in (Huang et al. 2016, DenseNet)  as vanilla SGD.
Therefore, the authors should consider more broader range of weight decay parameters for at least image classification tasks.

Minor:
1. In eq (2), the domain of x should be mentioned: according to Reddie et al, it is [-1,1].
2. In both theorem 4.1 and corollary 4.1, $D_{\infty^2}$ should be $D_{\infty}^2$?

**Experience Assessment:**

I have read many papers in this area.

**Review Assessment: Checking Correctness Of Derivations And Theory:**

I carefully checked the derivations and theory.

**Review Assessment: Checking Correctness Of Experiments:**

I carefully checked the experiments.

**Review Assessment: Thoroughness In Paper Reading:**

I read the paper at least twice and used my best judgement in assessing the paper.

---

> ### Author Response · Authors · 2019-11-06
> **Rebuttal: about the cons**
>
> Thank you for your valuable comments and inspiring questions. We want to clarify the following points about our new AdaX algorithm.
>
> $>>>$What is default hyperparameters for AdaX in Algorithm 1? Is it the same as AdaX-W (Algorithm 3) in Appendix?
>
>   Yes. The hyperparameters for AdaX are the same as in AdaX-W, i.e. $\beta_1=0.9, \beta_2=1e-4,\beta_3=0.999$. Actually, we mentioned in the Appendix of our paper that Algorithm 3 presented the "detailed implementations of AdaX and AdaX-W". The only difference between AdaX and AdaX-W is whether to use L-2 regularization or weight decay in the algorithm.
>
> $>>>$So, It is not clear that the second momentum estimate of Ada-X is really stable. For this, it would be interesting to see how the trajectories of second-order momentum estimates of Adam, AMSGrad, Ada-X are different.
>
>   The stability of the second momentum of AdaX is guaranteed by the convergence theorems (Theorem 4.1 and 4.2) in both the convex and the non-convex cases. In fact, the large bias-correction term would be scaled by the exponentially decreasing $1/\sqrt{(1+\beta_2)v_{t-1} + \beta_2 g_t^2}$ and hence the second momentum estimate of AdaX remains stable in the long term. However, we definitely agree that it would be interesting to see the different trajectories of second-order momentum estimates of different algorithms. We will add the trajectories of the different algorithms in our toy example (Problem (3)) to the final version of the paper. Thank you for such an interesting suggestion.
>
> $>>>$ Although the authors effectively remove log T in the numerator in Corollary 3.2 of Chen et al. (2019) using their lemma 4.1 (I think this is the key point), the assumption that $\beta_{2t} = \beta_2/t$ seems quite strong, and original Adam paper has no such assumptions.
>
>     Yes, you are right and the condition on $\beta_{2t}$ is quite strong. The original Adam paper doesn't have this strong assumption because it uses the exponential moving average of the square of the past gradients. Therefore, its second moment and bias correction will be bounded if the gradients are bounded. However, when we choose to employ exponential long-term memory, both terms will possibly go to infinity and such a condition is needed in the convergence proofs. We will consider improving this condition to $\beta_{2t} = \beta_2/\sqrt{t}$ in the future. Theorem 4.2 uses it to match with the settings of Theorem 4.1, and Corollary 4.2 actually shows that we don't need such a condition when analyzing convergence based on gradient sizes.
>
> $>>>$... In other words, the authors should use the same policy on constructing the first-order momentum estimate for both Adam and Ada-X. Also, as the authors add an additional hyperparameter $\beta_3$, the effect of $\beta_3$ on performance should be discussed at least empirically.
>
>     We emphasize that $(1-\beta_{3})$ helps to prove the convergence of AdaX and it only scales the step sizes by a constant. We can obtain the same strategy as Adam's first-order momentum by changing $(1-\beta_3)$ to $(1-\beta_1)$ and decreasing the learning rate by $(1-\beta_3)/(1-\beta_1)$. In such a case, the weight decay would naturally increase by $(1-\beta_1)/(1-\beta_3)$. We agree that we should add more discussions on the effect of $\beta_3$ in the experiments section to reveal this fact, and we will provide such experiments as soon as possible.
>
> $>>>$In this context, Zaheer et al. (2018, Adaptive methods for non-convex optimization) propose a large epsilon value (numerical stability parameter) such as for better generalization. It will be more interesting to see the comparisons in this regime.
>
>    Including a large epsilon is one way of fixing the poor generalization and we agree that such comparisons could be interesting to explore. Thank you for pointing it out. However, a large epsilon will also harm the adaptive ability of the algorithm since small gradients will be dominated by the large epsilon value in the second momentum and the algorithm will be similar to SGD. We will try running a few experiments in this direction if time allows.
>
> $>>>$In my experience with Adam-W (Decoupled weight decay regularization), Adam-W requires a relatively large weight decay parameter $\lambda$...
>
>   Thank you for your advice. However, we believe that this is just an implementation difference. As can be observed in Figure 1c, the higher weight decay 5e-4 already makes the AdamW algorithm converge as slow as SGDM. We also tried AdamW with higher and lower weight decays in our experiments, but they ended up with even worse results.
>
> $>>>$For the minor errors
>
>   Yes, you are absolutely right and thank you for being such a careful reader. We will certainly fix these issues in our next version of the paper.

---

### Decision · Program_Chairs · 2019-12-19

**Decision:**

Reject

**Comment:**

This paper analyzes the non-convergence issue in Adam in a simple non-convex case. The authors propose a new adaptive gradient descent algorithm based on exponential long term memory, and analyze its convergence in both convex and non-convex settings. The major weakness of this paper pointed out by many reviewers is its experimental evaluation, ranging from experimental design to missing comparison with strong baseline algorithms. I agree with the reviewers’ evaluation and thus recommend reject.